# Vegetation greenness and land carbon flux anomalies associated with climate variations: a focus on the year 2015

Chao Yue[1], Philippe Ciais[1], Ana Bastos[1], Frederic Chevallier[1], Yi Yin[1], Christian Rödenbeck[2], Taejin Park[3]

[1]Laboratoire des Sciences du Climat et de l'Environnement, CEA-CNRS-UVSQ, UMR8212, 91191 Gif-sur-Yvette, France
[2] Max Planck Institute for Biogeochemistry, Jena, Germany.
[3]Department of Earth and Environment, Boston University, Boston, MA 02215, USA

Corresponding author: Chao Yue, chao.yue@lsce.ipsl.fr

**Abstract**

Understanding the variations in global land carbon uptake, and their driving mechanisms, is essential if we are to predict future carbon cycle feedbacks on global environmental changes. Satellite observations of vegetation greenness have shown consistent greening across the globe over the past three decades. Such greening has driven the increasing land carbon sink, especially over the growing season in northern latitudes. On the other hand, interannual variations in land carbon uptake are strongly influenced by El Niño–Southern Oscillation (ENSO) climate variations. Marked reductions in land uptake and strong positive anomalies in the atmospheric $CO_2$ growth rates occur during El Niño events. Here we use the year 2015 as a natural experiment to examine the possible response of land ecosystems to a combination of vegetation greening and an El Niño event. The year 2015 was the greenest year since 2000 according to satellite observations, but a record atmospheric $CO_2$ growth rate also occurred due to a weaker than usual land carbon sink. Two atmospheric inversions indicate that the year 2015 had a higher than usual northern land carbon uptake in boreal spring and summer, consistent with the positive greening anomaly and strong warming. This strong uptake was, however, followed by a larger source of $CO_2$ in the autumn. For the year 2015, enhanced autumn carbon release clearly offset the extra uptake associated with greening during the summer. This finding leads us to speculate

that a long-term greening trend may foster more uptakes during the growing season, but no large
increase in annual carbon sequestration. For the tropics and Southern Hemisphere, a strong
transition towards a large carbon source for the last three months of 2015 is discovered,
concomitant with El Niño development. This transition of terrestrial tropical $CO_2$ fluxes between
two consecutive seasons is the largest ever found in the inversion records. The strong transition
to a carbon source in the tropics with the peak of El Niño is consistent with historical
observations, but the detailed mechanisms underlying such extreme transition remain to be
elucidated.

**1 Introduction**

The first monitoring station for background atmospheric $CO_2$ concentration was established on
Mauna Loa in 1958. Its record shows that atmospheric $CO_2$ has continued to rise as
anthropogenic carbon emissions have increased. However, annual atmospheric $CO_2$ growth rates
(AGR) are lower than those implied by anthropogenic emissions alone, because land ecosystems
and the oceans have absorbed part of the emitted $CO_2$ (Canadell et al., 2007; Le Quéré et al.,
2016). At multi-decadal timescale, carbon uptake by land and ocean has kept pace with growing
carbon emissions (Ballantyne et al., 2012; Li et al., 2016), exerting a strong negative feedback to
global change. Over land, increasing carbon uptake is consistent with worldwide vegetation
greening as revealed by satellite observations (Zhu et al., 2016). Long-term warming and $CO_2$
fertilization have contributed to growing-season greening and increasing land carbon uptake in
northern latitudes, which further leads to a markedly increasing seasonal atmospheric $CO_2$
amplitude (Forkel et al., 2016; Graven et al., 2013; Myneni et al., 1997). At the same time, large
year-to-year fluctuations occur in the terrestrial carbon sink especially over tropical lands. These
fluctuations mainly occur in response to climate variations induced by El Niño–Southern
Oscillation (ENSO) (Wang et al., 2013, 2014) and other occasional events such as volcanic
eruptions (Gu et al., 2003). The occurrence of El Niño events often leads to elevated
temperatures with reductions in precipitation over tropical lands; these cause sustained droughts
that substantially reduce the land carbon uptake (Doughty et al., 2015). These ENSO-associated
tropical land uptake variations have translated into large variations in atmospheric $CO_2$ growth
rates, which are found to be significantly correlated with tropical land temperature anomalies
(Wang et al., 2014).

Understanding such driving mechanisms and variations in land carbon uptake is essential if we
are to predict future carbon cycle feedbacks on global environmental changes — including
climate change. Growing-season normalized difference vegetation index (NDVI) observed by
the moderate-resolution imaging spectroradiometer (MODIS) aboard the Terra satellite has
consistently increased since 2000 over northern latitudes (Bastos et al., 2017). The positive link
between seasonal NDVI and vegetation photosynthesis is well established for deciduous forests
in temperate and boreal biomes (Gamon et al., 1995), and NDVI is often assumed as a surrogate
for vegetation growth. Nevertheless, it is unclear whether, on an annual timescale, higher NDVI
anomalies are indeed always associated with higher carbon uptake anomalies. A few studies have
reported important seasonal coupling in vegetation greenness and land carbon uptake in mid to
high latitudes, with summer drought likely compensating for enhanced spring uptake (Angert et
al., 2005; Wolf et al., 2016). While spring warming may enhance vegetation growth and carbon
uptake, autumn warming can lead to net carbon loss by enhancing respiration carbon loss (Piao
et al., 2008). Furthermore, greening and browning may occur in different regions within the same
growing season (Bastos et al., 2017), so that the associated consequence on land carbon
dynamics needs to be investigated. Such investigation on the relationship between seasonal
NDVI dynamics and land carbon uptake will help to predict future land carbon sink capacity.

El Niño events are often linked to enhanced drought conditions in Amazonian forest with
widespread increases in tree mortality and drops in ecosystem carbon storage (Phillips et al.,
2009). Fire emissions from Asian tropical regions also show nonlinear increase with drought
during El Niño, another factor that reduces land carbon sequestration (Field et al., 2016; Yin et
al., 2016). With future anthropogenic climate change, it is projected that the frequency of
extreme El Niño events will be doubled (Cai et al., 2014). It is not clear how future carbon
dynamics will respond to such a doubling of extreme El Niño events, or whether any extreme
phenomena in the carbon cycle might occur as a consequence. The year of 2015 is a good test
case for investigating El Niño-related phenomena because it contained the strongest El Niño
event since the one in 1997/98 and occurred at a time when unprecedented mean annual land
temperature was observed.

We use the year 2015 as a natural experiment to investigate the response of land ecosystems to a combination of extreme greening and an El Niño event. The year 2015 had the greenest growing season in the Northern Hemisphere since 2000, in particular over eastern North America and large parts of Siberia (Bastos et al., 2017), and this was accompanied by the highest mean annual global land temperature on record since 1880 (https://www.ncdc.noaa.gov/cag/time-series/global/globe/land/ytd/12/1880-2015). At the same time, a strong El Niño event developed starting in the latter half of 2015. Elevated fire emissions in tropical Asia were reported (Huijnen et al., 2016; Yin et al., 2016) and a severe drought was detected over eastern Amazonia (Jiménez-Muñoz et al., 2016). As a result, in 2015 the global monthly atmospheric $CO_2$ concentration surpassed 400 $\mu mol \cdot mol^{-1}$ (ppm) for the first time, with an unprecedented large annual growth rate of 2.96±0.09 ppm $yr^{-1}$ (https://www.esrl.noaa.gov/gmd/ccgg/trends/global.html#global_growth). We examined global and regional land-atmosphere carbon fluxes estimated from two atmospheric inversions over 1981-2015. Seasonal patterns in the land carbon uptake in 2015 relative to the long-term trend of 1981-2015 were examined and linked to extreme climate anomalies. To put the 2015 land response into a historical context, we also examined the relationship between historical land carbon uptake anomalies, and NDVI and climate anomalies, in order to infer general patterns in factors driving the land carbon uptake anomalies. Our findings are expected to provide insights into future land carbon cycle feedbacks to vegetation greening and climate variations.

**2 Data and methods**

**2.1 Data sets**

**2.1.1 Atmospheric inversion data**

We used two gridded land and ocean carbon uptake data sets based on atmospheric $CO_2$ observations, namely those from the Copernicus Atmosphere Monitoring Service (CAMS) inversion system developed at LSCE (Chevallier et al., 2005, 2010) and the Jena CarboScope inversion system developed at the MPI for Biogeochemistry, Jena (update of Rödenbeck, 2005; Rödenbeck et al., 2003). Atmospheric inversions estimate land- and ocean-atmosphere net carbon fluxes by minimizing a Bayesian cost function, accounting for the mismatch between the observed and simulated atmospheric $CO_2$ mixing ratios. To do this, they use atmospheric $CO_2$

concentrations at measurement sites, combined with an atmospheric transport model and prior
information on fossil fuel carbon emissions and carbon exchange between the atmosphere and
land (and ocean). Detailed information on these two atmospheric inversions can be found in the
references above.

The CAMS inversion data (version r15v3) were provided for 1979–2015 with a weekly time-step
and a spatial resolution of 1.875° latitude and 3.75° longitude. The Jena CarboScope inversion
provides daily fluxes at a spatial resolution of 3.75° latitude and 5° longitude and offers a series
of runs. All runs provide data covering the whole period of 1979–2015, but they have different
validity periods that focus on using different sets of $CO_2$ measurement stations. Within the
validity period all employed stations have valid $CO_2$ observations, i.e., they have coherent and
complete measurements over time. The idea of validity period is to avoid spurious flux variations
resulting from a changing station network. It is optimal to examine temporal trend within the
validity period, but this does not mean the data outside this period are invalid and should be
discarded. From the Jena inversion runs, we selected s04_v3.8 (shortened as Jena04 in the main
text and the Supplementary Material). Jena04 used the largest number of measurement sites for
2015 and therefore had the most detailed constraint on carbon exchanges for this year (see
http://www.bgc-jena.mpg.de/CarboScope/ for more details on other configurations). The validity
period of the Jena04 data is 2004–2015, but here we used the data available for the whole time
span of 1981–2015. This is necessary to provide both the large number of sites in 2015 and the
long historical period, which is needed to produce a robust anomaly estimate. We compared the
linear trends obtained over large latitudinal regions between the Jena04 run and the long
s81_v3.8 run (the latter has a validity period of 1981–2015 but far fewer sites are included than
in Jena04), and confirmed that the derived trends are similar. The same issue of evolving site
number and data coverage with time also occurs for the CAMS inversion, but the CAMS
inversion uses sites with at least a 5-year run of data. CAMS therefore has a denser (during the
recent decade) but temporally evolving data coverage than Carboscope.

Estimates of land and ocean net carbon uptakes for 1981–2015 from the Global Carbon Project
(GCP) (Le Quéré et al., 2016) were compared with the inversion data. For this purpose, an
annual global carbon flux of 0.45 Pg C yr$^{-1}$ is subtracted from the inversion-derived land carbon
uptakes and is added to ocean carbon uptakes to account for the pre-industrial land-to-ocean
carbon fluxes induced by river transport (Jacobson et al., 2007), following Le Quéré et al. (2016).
Ocean carbon uptakes in the GCP estimates are based on the mean $CO_2$ sink estimated for the
1990s from observations, and the trend and variability in the ocean $CO_2$ sink for 1959–2015 from
global ocean biogeochemistry models. Estimates of land carbon uptake in the GCP estimates are
calculated as the difference between anthropogenic emissions, atmospheric $CO_2$ growth and the
ocean sink. In summary, the estimates of land and ocean carbon uptake in the GCP estimates are
largely independent from the two inversions used here, except that the $CO_2$ records from
atmospheric stations that are used in the inversions are also used in the GCP estimates.

**2.1.2 Atmospheric $CO_2$ growth rates, NDVI and climate data**
Atmospheric $CO_2$ growth rates were retrieved from the Global Monitoring Division, Earth
System                Research                Laboratory                (ESRL),                NOAA
(http://www.esrl.noaa.gov/gmd/ccgg/trends/global.html). We used NDVI data between 2000 and
2015 from MODIS Terra Collection 6 (Didan, 2015), at a resolution of 0.05° and a 16-day time-
step. NDVI data are processed from MODIS land surface reflectance data and thoroughly
corrected for atmospheric effects. We were strict in applying quality assurance (QA) controls to
maintain a distinct seasonal trajectory of vegetative radiometric observations and minimize
spurious signals (e.g., snow or cloud). Detected unexpected non-vegetative observations were
first excluded and then filled using the adaptive Savitzky–Golay filter (Chen et al., 2004; Jönsson
and Eklundh, 2004). The Savitzky–Golay filter is a simplified convolution over a set of
consecutive values with weighting coefficients given by a polynomial least-squares-fit within the
filter window (Savitzky and Golay, 1964). After this procedure, the linearly interpolated daily
NDVI data were used to calculate mean seasonal NDVI and re-gridded at 0.5° resolution, with
pixels of seasonal NDVI lower than 0.1 being further masked to ensure robustness. We examined
four seasons: Q1 (January–March), Q2 (April–June), Q3 (July–September) and Q4 (October–
December). Climate fields are from the ERA interim reanalysis (Dee et al., 2011) at 0.5°
resolution and monthly time-step. We used air temperature, precipitation and volumetric soil
water content (%) integrated over the soil column to a depth of 2.89 m.

**2.1.3 Indices for El Niño–Southern Oscillation states and fire emission data**
We examined the seasonal variations of the carbon cycle in 2015 in relation to ENSO events and
compared the 2015 El Niño event with that of 1997–1998. The Multivariate ENSO Index (MEI,
http://www.esrl.noaa.gov/psd/enso/mei/, Wolter and Timlin, 2011) was used to indicate the
ENSO state. MEI is a composite index calculated as the first un-rotated principal component of
six ENSO-relevant variables (including sea level pressure and sea-surface temperature) over the
tropical Pacific for each of twelve sliding two-monthly seasons. MEI has been widely used in
previous studies of land carbon dynamics as an indicator of ENSO states (Nemani et al., 2003;
van der Werf et al., 2008). The twelve two-monthly MEI values for each year are summed to
obtain the annual MEI. The interannual variations in climate and land carbon uptake are linked
with MEI to infer a general relationship between land carbon dynamics and ENSO climate
oscillations. To examine the potential role of fire emissions in the land carbon balance in 2015,
we used the GFED4s carbon emission data at daily time-step and 0.25° spatial resolution
(http://www.globalfiredata.org/data.html). Monthly fire-carbon emissions were calculated for the
regions and were examined for 1997–2015.

**2.2 Data analysis**
**2.2.1 NDVI rank analysis and greening trend**
We first examine the vegetation greenness status in the year 2015. Given a season and a pixel,
the annual time series of seasonal NDVI for 2000-2015 were ranked in ascending order so that
each year could be labelled by a rank, with 1 being the lowest and 16 being the highest. A spatial
map of NDVI rank was then obtained for each year for the given season (Fig. S1). A composite
map was made for 2015, by merging pixels with the highest rank of all four seasons in 2015 (Fig.
1a). Vegetated area fraction with the highest rank for different years was obtained, with the sum
of these fractions yielding unity. This procedure was repeated for all four seasons to generate
four seasonal time series, with each time series containing the vegetation land fractions with
highest NDVI for different years (Fig. 1b). Note that NDVI values for the Northern Hemisphere
for Q1 and Q4 mostly fall outside the growing season (although October is frequently considered
within the growing season and some evergreen coniferous forests show significant
photosynthetic activities in March in regions with mild winters, e.g., Tanja et al., 2003), so that a
valid NDVI might not necessarily be associated with significant seasonal vegetation activity.
However, we expect that this issue will be partly alleviated by our applied rigorous QA control
in preprocessing and a minimum threshold of 0.1 on seasonal NDVI. Such seasonal segregation
is adopted mainly because of its general applicability across the globe, especially for tropical
ecosystems where seasonality in vegetation activities is minimal.

**2.2.2 Analysis of land carbon uptake dynamics associated with climate variations**

Annual land and ocean carbon uptake and carbon emissions from the two inversions were
calculated for the globe over their period of overlap, 1981–2015. AGRs from NOAA/ESRL over
1981–2015 were converted into Pg C using a conversion factor of 2.12 Pg C ppm$^{-1}$ (Ballantyne
et al., 2012; Prather et al., 2012; Quéré et al., 2016), in order to examine the closure of the global
carbon balance in the inversion data. The conversion factor used here assumes that the entire
atmosphere is well mixed over one year. We attributed the record high AGR in 2015 into
individual components of emissions and sinks. The record high AGR in 2015 was a composite
effect collectively determined by carbon emissions from fossil fuel burning and industry, and
land and ocean carbon uptakes. All these were impacted by an historical trend (Fig. 2).
Therefore, to understand the factors contributing to 2015's record AGR, we separated it into a
long-term trend and interannual anomalies. Annual time series of carbon emissions, land and
ocean carbon uptakes, and AGRs from NOAA/ESRL over 1981-2015 were linearly de-trended.
The percentages of anomalies in carbon emissions, land and ocean sink in 2015 to the 2015 AGR
anomaly were then calculated as relative contributions by each factor to the 2015 AGR anomaly.

Seasonal land carbon uptake anomaly time series were also calculated by subtracting the same
linear trend for 1981–2015. The globe was divided into three latitude bands: boreal Northern
Hemisphere (BoNH, latitude > 45°N), temperate Northern Hemisphere (TeNH, 23.5° < latitude
< 45°N), and tropics and extratropical Southern Hemisphere (TroSH, latitude < 23.5°N). The
BoNH and TeNH are grouped as Boreal and temperate Northern Hemisphere (BoTeNH, latitude >
23.5°N) when examining seasonal carbon transitions. Seasonal land carbon uptake anomalies
were then calculated for each region and the whole globe, with positive anomalies indicating
enhanced sink (or reduced source) against the linear trend (i.e., the normal state), and negative
ones indicating the opposite. The same seasonal linear de-trending was also performed for
climate fields of air temperature, precipitation and soil water content. The relationship between
anomalies in land carbon uptake, and temperature and precipitation are examined using partial
correlation coefficients in a multivariate linear regression framework with an ordinary least
squares method. The relationships between seasonal land uptake anomalies and NDVI anomalies
were also examined using simple linear regression.

We then examined especially the seasonal anomalies of land carbon uptake in 2015 and the
carbon uptake transitions between two consecutive seasons, to reveal any extreme phenomena in
the land carbon cycle that might lead to the abnormally high AGR in 2015. Seasonal land carbon
uptake transitions are calculated as the land sink anomaly in a given season minus that of the
previous one. When examining transitions of land carbon uptake anomalies by the CAMS
inversion, we found the year 1993 had an extreme negative Q3→Q4 global transition (-2.85 Pg
C within 6 months, < -4σ, the second lowest being the year 2015 with -1.0 Pg C) albeit with a
reasonable annual land carbon uptake (3.75 Pg C $yr^{-1}$). This is linked to an extreme high Q3 and
low Q4 uptake in this year, which could not be explained by any known carbon cycle
mechanisms. This is thus identified as a result of numerical instability of the inversion system for
that release and consequently the year 1993 has been removed from all the aforementioned
seasonal analyses. However, we identified that the temporal trends for annual and seasonal land
carbon uptakes show almost no change whether or not including the year 1993.

**3 Results**
**3.1 Vegetation greening in 2015**
Figure 1a illustrates where and when higher-than-normal greening conditions were observed in
different seasons of the year 2015, compared to other years of 2000–2015 (see Supplementary
Material Fig. S1 for the greenness distribution of each season). On average over the four seasons
of 2015, 16% of vegetated land shows record seasonal NDVI.  The year with the second highest
NDVI is 2014 with 9% vegetated area having record NDVI. An increase of the record-breaking
NDVI occurrence over time is clearly seen in Fig. 1b. In short, 2015 clearly stands out as a
greening outlier, having the highest proportion of vegetated land being the greenest for all four
seasons except for the first season (despite the fact that for Q1, 2015 is still the third highest, Q1
= January to March).

For boreal and temperate regions of the Northern Hemisphere, the seasons with highest NDVI in

2015 are dominated by Q2 and Q3 (Q2 = April to June; Q3 = July to September), corresponding to the growing season from spring to early autumn (Supplementary Material Fig. S2). A pronounced greening anomaly in Q2 occurred in western to central Siberia, western Canada and Alaska, and eastern and southern Asia (Supplementary Material Fig. S1). Central and eastern Siberia and eastern North America showed marked greening in Q3. Strong and widespread greening also occurred in the tropics during Q3 across Amazonia and the savanna (or cerrado) of eastern South America and tropical Africa, but this strong positive greening signal greatly diminished in Q4 (Q4 = October to December) especially in central to eastern Amazonia with the development of El Niño (Supplementary Material Fig. S1). Overall, the strongest greening in 2015 across the globe is dominated by northern lands (latitude > 23.5°N), while for the northern tropics (0–23.5°N) only moderately strong greening is found, and for the Southern Hemisphere the greening of 2015 is close to the average state for the period of 2000–2015 (Supplementary Material Fig. S3). The extreme growing-season greening in the northern land is confirmed as being a robust result by Bastos et al. (2017), who used Terra MODIS NDVI data with different quality control procedures, and consistency is also confirmed between Terra and Aqua sensors (Fig. S1 in Bastos et al., 2017).

**3.2 Global carbon balance for 1981-2015**

Figure 2 shows the time series of fossil fuel burning and industry carbon emissions, NOAA/ESRL AGR rates linked to ENSO climate oscillations as indicated by the Multivariate ENSO Index (MEI), and land and ocean carbon sinks for the common period of the two inversions (1981–2015) and the estimates by the Global Carbon Project (GCP). Emissions show a clear increase with time, however AGRs are more variable. The record high AGR of 2.96 ppm in 2015 exceeds those in all previous years including the extreme El Niño event in 1997–98, despite much higher annual emissions in 2015. Interannual variability in AGR is mainly caused by fluctuations in the land carbon sink, with Pearson's correlation coefficients between de-trended AGR and land sink < -0.8 (p<0.01) for both inversions (Pearson's correlation coefficient between de-trended AGR and MEI being 0.27, p<0.1). The root mean square differences between inversion and GCP carbon sinks are 0.70 and 0.65 Pg C yr$^{-1}$ for CAMS and Jena04 respectively for the land, and ~0.5 Pg C yr$^{-1}$ for the ocean for both inversions, within the uncertainties of 0.8 and 0.5 Pg C yr$^{-1}$ over 1981–2015, respectively for land and ocean as

reported by GCP. The interannual variability of de-trended sink anomalies for the land agrees
well between the inversions and the GCP estimates (with Pearson's correlation coefficient being
0.9 for both inversions, p < 0.01).

For 2015, the prescribed anthropogenic carbon emissions in the CAMS inversion are 9.9 Pg C yr$^{-}$
$^{1}$, of which 2.0 Pg C are absorbed by ocean, 1.7 Pg C by land ecosystems, with 6.2 Pg C
remaining in the atmosphere, which matches the AGR from background stations of 6.3 Pg C
assuming a conversion factor of 2.12 Pg C ppm$^{-1}$ (Ballantyne et al., 2012; Le Quéré et al., 2016)
and considering a measurement uncertainty in AGR of 0.09 ppm (0.2 Pg C) for 2015. When land
carbon fluxes from the inversion are linearly de-trended over 1981-2015, the terrestrial sink in
2015 is 1.2 Pg C lower than normal (i.e., the trend value), but this is not an extreme value — it is
only the seventh weakest sink since 1981. This weaker land uptake accounts for 82% of the
positive AGR anomaly, which is 1.45 Pg C in 2015 by subtracting a linear temporal trend.
Jena04 yields an AGR in 2015 that is 0.13 ppm lower than the AGR based on background
stations only, a difference close to the observation uncertainty. After removing the linear trends
over time similarly as for the CAMS inversion, the land carbon uptake anomaly for Jena04 is -
0.3 Pg C yr$^{-1}$ in 2015, or 20% of the observed AGR anomaly, the remaining being explained by a
positive anomaly in fossil fuel emissions (34%), a negative anomaly in the ocean sink (20%),
and the difference between modelled AGR and NOAA/ESRL reported AGR. Note that the land
sink given by the GCP data for 2015 is much lower than in the two inversions, with de-trended
anomaly lower than that of CAMS, indicating an even larger contribution from land to the high
anomaly of AGR.

In general, the warm phases of ENSO events are associated with positive anomalies in land air
temperature, negative precipitation anomalies, and lower land carbon uptake anomalies (Fig. 3),
this is consistent with previous studies (Cox et al., 2013; Wang et al., 2014). The lower
precipitation during El Niño is due to a shift of precipitation from tropical land to the ocean
(Adler et al., 2003), and higher land temperature might be due to reduced evaporative cooling.
The two extreme El Niño years of 1997 and 2015 have rather close MEI values. Compared with
the 'standard' El Niño state of temperature and precipitation represented by the regression line,
the year 1997 was relatively 'cool' and 'wet', while 2015 was rather 'warm' and 'dry' (with an
extremely negative precipitation anomaly). Year 1998 has a smaller value of MEI than
1997/2015, but has a higher temperature anomaly than 2015, and a much lower land carbon
uptake anomaly than 1997 and 2015 in both inversions, while the land carbon uptake anomalies
in 1997 and 2015 are similar. More detailed comparison of these three years and their carbon
cycle dynamics will be presented in the discussion section.

**3.3 Seasonal land carbon uptake dynamics associated with climate variations with a focus**
**on 2015**

Figure 4 shows the partial correlation coefficients between anomalies in seasonal land carbon
uptake and those in seasonal temperature and precipitation for different regions. The simple,
individual (univariate) linear relationships between de-trended anomalies in land carbon fluxes
and those in temperature and precipitation, are presented in the Supplementary Material (Fig. S4
and S5). Land carbon fluxes show consistent relationships with temperature between the two
inversions for BoNH: a positive relationship for Q2 and a negative one for the other three
seasons (with Q1 by Jena04 being the only one with a non-significant correlation). Partial
correlations between land fluxes and precipitation are absent or non-significant for BoNH. This
points to the fact that vegetation productivity in BoNH is in principle dominated by temperature,
with warmer spring and early summer (Q2, April–June) enhancing vegetation net carbon uptake,
but a higher temperature in later summer, autumn and early winter reducing the land capacity to
sequester carbon, consistent with previous studies (Piao et al., 2008). For TeNH, a significant
negative relationship is found between land fluxes by the CAMS inversion and temperature for
Q3, and both inversions show negative relationships between land fluxes and precipitation for
Q4, probably due to enhanced early autumn respiration under wetter conditions. For TroSH, land
carbon uptakes in Q1, Q2 and Q4 are all negatively related with temperature ($p < 0.05$ for both
inversions), while increase in precipitation in Q1 is found to be associated with enhanced land
uptake.

To explain the apparent paradox in 2015 between the strong greening and an only moderate
terrestrial uptake, we examined in detail the seasonal land carbon flux anomalies in 2015 (Fig. 5,
refer to Supplementary Material Fig. S6 for the spatial distribution of flux anomalies). At
seasonal scale, both inversions indicate positive carbon uptake anomalies during Q2 and Q3 for
boreal and temperate Northern Hemisphere (BoTeNH, latitude > 23.5°N), consistent with
marked greening in central to eastern Siberia, eastern Europe and Canada (Fig. 1) as outlined
above. Indeed, both BoNH and TeNH show positive relationships between seasonal land carbon
flux anomalies and NDVI anomalies for Q2 and Q3, with BoNH showing moderate greenness
(after a linear trend is removed) for Q3, and TeNH showing extreme greenness for Q2 in 2015
(Supplementary Material Fig. S7). However, an extreme follow-up negative (source) anomaly
occurred in Q4 (Fig. 5a). These negative anomalies were lower than the 10th percentile of all
anomalies in Q4 over time for both inversions and they partly cancelled the extra uptake in Q2
and Q3. As a result, on an annual timescale, the CAMS inversion shows an almost neutral land
flux anomaly in BoTeNH, while the Jena04 inversion still indicates a significant positive annual
anomaly.

For the tropics and extratropical Southern Hemisphere (TroSH, latitude < 23.5°N), both
inversions show a weak negative land carbon anomaly for Q1 (mean value of -0.10 Pg C) in
2015, moderate anomalies in Q2 (of differing signs, with a negative one of -0.3 Pg C in CAMS
and a positive one of 0.2 Pg C in Jena04). Q3 anomalies are almost carbon neutral for both
inversions. In stark contrast, between Q3 and Q4, both inversions show a strong shift towards an
abnormally big land carbon source (i.e., negative anomalies of ~ -0.7 Pg C against a carbon
source expected from the linear trend, lower than 10th percentile over time in both inversions).
On an annual timescale, CAMS shows a large negative anomaly of -1.2 Pg C. For Jena04, sink
and source effects in Q1–Q3 cancelled each other, leaving the annual anomaly the same as in Q4.

Over the globe, the Jena04 inversion shows an abnormally strong sink during Q2 (normal state
being a net carbon sink), owing to synergy of enhanced Q2 uptakes in both BoTeNH and TroSH.
This abnormally enhanced uptake partly counteracted the strong shift towards a source in Q4
(normal state being a net carbon source), leaving a small negative annual land carbon balance of
-0.3 Pg C. For the CAMS inversion, because of the co-occurrence of enhanced carbon release in
BoTeNH and the sudden shift towards a large carbon source in TroSH both in Q4 (normal state
for both being a net carbon source), the land shows a strong global shift towards being a source
in Q4, leaving a negative annual carbon anomaly of -1.2 Pg C (i.e., carbon sink being reduced
compared with the normal state).

These consistent results from both inversions point to very strong seasonal shifts in the land
carbon balance as an emerging feature of 2015. We thus calculated ***transitions*** in land carbon
uptake anomaly as the difference in flux anomalies between two consecutive seasons (defined as
the anomaly in a given season minus that in the previous one) for all years of the period 1982-
2015 (Fig. 6). The ranks of transitions for different seasons relative to other years between the
two inversions are broadly similar, except for Q1→Q2 and Q2→Q3 in TroSH, mainly due to the
differences between the two inversions in seasonal land-carbon uptake anomaly in Q2 (Fig. 5b).
On the global scale, both inversions show an extreme transition to a negative uptake anomaly for
Q3→Q4, with 2015 being the largest transition of the period 1982-2015 (a transition towards an
enhanced carbon source of -1.0 Pg C in six months). The abnormal transitions for Q3→Q4 on
the global scale are located in the TroSH region, where both inversions show that during 1982-
2015 the largest transition occurred in 2015. For BoTeNH, both inversions showed strong
transitions towards positive anomaly for Q1→Q2; however, the same strong transition towards
source anomaly occurred in Q3→Q4, partly cancelling the sink effects during growing seasons.

**4 Discussion**
**4.1 Land carbon uptake dynamics with climate variations in northern latitudes and**
**seasonal transitions of land carbon uptake in 2015**
The two inversions consistently allocate a strong positive carbon uptake anomaly in the region of
BoTeNH during spring, which persists through the summer (Q2–Q3): an extreme sink anomaly
is estimated in Q2 by Jena04, but a more moderate one by CAMS (still above the 75th
percentile). The strong sinks in Q2 in both inversions are dominated by temperate Northern
Hemisphere regions (TeNH, 23.5° < latitude < 45°N, Supplementary Material Fig. S8). For this
region, both inversions show strong positive correlation between carbon uptake anomalies and
NDVI in Q2, with an extremely high NDVI anomaly in 2015 (Supplementary Material Fig. S7f).
Thus, the strong sinks in Q2 are evidently linked to the extreme greening, while temperature and
precipitation anomalies were only moderate (Fig. S4f, Fig. S5f).

For Q3, an extreme carbon sink anomaly occurs in boreal Northern Hemisphere (BoNH, latitude >
45°N) in CAMS; however, an equally strong negative anomaly (i.e., reduced sink) was found in
TeNH in the same season, leaving the whole boreal and temperate Northern Hemisphere
(BoTeNH) only a moderately positive sink anomaly (Fig. S8). For TeNH alone, CAMS indicates
extreme seasonal shift from a positive anomaly in Q2 to a negative one in Q3. This implies
strong seasonal transitions resulting from enhanced ecosystem $CO_2$ release after growing-season
uptake and the presence of seasonal coupling in land carbon dynamics. For TeNH in 2015,
NDVI persisted from a high extreme in Q2 to close to normal in Q3 (Fig. S7f, Fig. S7g), and
temperature remained moderate for both Q2 and Q3 (Fig. S4f, S4g), but precipitation shifted
from a moderate anomaly in Q2 to an extremely low one (Fig. S5f, S5g). In summary, the shift
from a high Q2 sink anomaly to a big Q3 source anomaly by CAMS might be linked to the shift
in precipitation and drought in Q3, such as the prevailing drought in Europe as shown in Fig. S9
(see also a detailed discussion of the European drought by Orth et al., 2016).

Jena04 inversion agrees with a higher-than-normal sink in TeNH (23.5° < latitude < 45°N)
during spring (Q2). It also reports a moderate positive anomaly for Q3 in BoNH, but does not
show a strong negative anomaly (i.e., reduced sink) in TeNH in Q3 as CAMS does (Fig. S8).
This is possibly related to differences in the measurement station data used, to different land
prior fluxes (from the ORCHIDEE model in CAMS, and the LPJ model in Jena CarboScope), or
to the fact that the Jena04 inversion has a larger a-priori spatial error correlation length for its
land fluxes (1275 km) than CAMS (500 km) (Chevallier et al., 2010; Rödenbeck et al., 2003).
Nonetheless, both inversions consistently indicate that the enhancement of $CO_2$ uptake during
spring and summer at the northern hemispheric scale was subsequently offset by an extreme
source anomaly in autumn (Q4).

The large carbon source anomalies in Q4 shown by the two inversions in BoTeNH seem to be
dominated by different factors in BoNH versus TeNH. In BoNH the source anomaly in 2015 is
more linked to elevated temperature in Q4, which shows a significant negative correlation with
carbon uptake anomalies by both inversions (Fig. S4d). In contrast, precipitation in Q4 has no
correlation with carbon uptake anomalies, and precipitation in 2015 was close to the normal state
(Fig. S5d). The prevailing high temperature in Q4 of 2015 is especially evident over most
northern America, and central to eastern Siberia and Europe (Supplementary Fig. S9a).

In TeNH, the roles of temperature and precipitation are reversed compared to BoNH. Q4 precipitation is found to have a significant negative correlation with land carbon uptake anomalies for both inversions, and Q4 in 2015 was characterized by a very high precipitation anomaly, leading to a reduced land carbon uptake (Fig. S5h). While temperature in Q4 of 2015 was moderately high, no significant correlation is found between carbon uptake anomalies and temperature (Fig. S4h). However, for both BoNH and TeNH, NDVI remained moderately high in Q4 of 2015 (Fig. S7d, S7h).

The positive relationship between land carbon uptake and temperature in Q2 (spring and early summer), and a negative one for Q3 and Q4 (autumn) for BoNH, are in line with previous studies. Several studies reported an enhanced greening during spring and summer in the Northern Hemisphere (Myneni et al., 1997; Zhou et al., 2001), as driven by increasing spring and summer temperatures (Barichivich et al., 2013; Nemani et al., 2003), leading to enhanced land carbon uptake and a long-term increase in the seasonal amplitude of atmospheric $CO_2$ in northern latitudes (Forkel et al., 2016; Graven et al., 2013). However, for autumn, even though ending of growing season has been delayed because of autumn warming (Barichivich et al., 2013), land carbon uptake termination time is found to have advanced as well, due to enhanced autumn respiration (Piao et al., 2008), which ultimately reduced the annual net ecosystem carbon uptake (Hadden and Grelle, 2016; Ueyama et al., 2014). For TeNH, we also found a significant negative relationship between land carbon uptake anomalies and temperature for Q3 using the CAMS inversion data, consistent with the enhanced respiration by autumn warming found in the aforementioned studies. For Q4, however, both inversions point to decreasing land carbon uptakes with increasing precipitation. This finding might be due to enhanced respiration resulting from higher soil moisture content, but further site-scale examination is needed to confirm this hypothesis.

For BoNH and TeNH, land carbon uptake anomalies are closely coupled with NDVI anomalies for Q2 (positive correlation, albeit an insignificant one for TeNH Q2 using Jena04 data), but they are generally de-coupled for Q3 and Q4, except that for Q3 of BoNH the CAMS-based land carbon uptake shows positive correlation with NDVI. This suggests high NDVI in autumn might

not necessarily relate to a high land carbon uptake. There are two reasons. First, NDVI is found
to correlate well with leaf-level $CO_2$ uptake for deciduous forest for different seasons, but is
largely independent of leaf photosynthesis for evergreen forests (Gamon et al., 1995). Second,
even though a higher NDVI is associated with larger photosynthetic capacity and a higher gross
photosynthesis, autumn warming might increase ecosystem respiration more than photosynthesis,
leaving a net carbon source effect. Furthermore, other studies have also pointed out that severe
summer drought can negate the enhanced carbon uptake during warm springs (Angert et al.,
2005; Wolf et al., 2016).

**4.2 Seasonal land carbon uptake transitions in the tropics and influences of El Niño and**
**vegetation fire**


The strong transition to abnormal source in the tropics and extratropical Southern Hemisphere
was paralleled by a marked decrease in precipitation and an increase in temperature in Q4, with
the development of El Niño in Q2–Q3 (Supplementary Material Fig. S4l, S5l, S10). Here El
Niño development is indicated by the rise of the MEI and Oceanic Niño Index (ONI,
http://www.cpc.ncep.noaa.gov/products/analysis_monitoring/ensostuff/ONI_change.shtml). This
strong transition is consistent with the expected response of tropical and sub-tropical southern
ecosystems during previous El Niño events (Ahlström et al., 2015; Cox et al., 2013; Poulter et
al., 2014; Wang et al., 2013, 2014). The small abnormal source in Q1 in TroSH is consistent with
a low precipitation anomaly. While temperature anomalies are abnormally high in Q2 and Q3,
accompanied by extremely negative precipitation anomalies, the extremely low carbon flux in
Q4 is largely explained by temperature, because correlations between land carbon uptake and
precipitation in Q4 are very weak (Fig. S4i–l, Fig. S5i–l). Vegetation greenness has significant
positive correlation with land carbon uptake anomalies only for Q1 in the tropics, and for the rest
of the three seasons the correlation is very weak (Fig. S7i–l).

Compared with the 1997–98 El Niño, which had a slightly larger MEI value, the 2015 El Niño
started much earlier, with positive MEI and ONI appearing during the first half of 2014. Since
then and until Q3 and Q4 in 2015 when El Niño began to reach its peak, the tropics and Southern
Hemisphere saw continuous higher-than-normal temperatures, with continually decreasing
precipitation and accumulating deficit in soil water content (Supplementary Material Fig. S10).
From Q3 to Q4, a steep decline is further observed in both precipitation and soil moisture with
stagnating high temperature anomaly, which is probably a major cause of the strong shift
towards a carbon source anomaly. The CAMS inversion shows a carbon source anomaly in Q4
of 2015 slightly smaller than that in Q3 of 1997, while the Jena04 inversion shows almost equal
magnitudes of loss in land sink strength between these two extreme El Niño events. On the one
hand, El Niño in late 2015 started early and built upon the cumulative effects of the drought
since the beginning of the year; it thus came with larger negative anomaly in precipitation and
soil water content than the 1997–98 El Niño. This sequence of events might favour a stronger
land carbon source. On the other hand, the fire emission anomaly in the tropics in 2015 was less
than half of that in 1997 at the peak of El Niño (Fig. S10); this might have contributed to a
smaller land source anomaly in 2015 than in 1997–98.

El Niño events are usually associated with increased vegetation fires, which have a large impact
on the global carbon cycle (van der Werf et al., 2004). Global fire emissions of carbon reached
3.0 and 2.9 Pg C in 1997 and 1998 according to the GFED4s data. These two years produced the
largest source of fire-emitted carbon for the entire period 1997–2015. In comparison, global fire
emissions in 2015 reached 2.3 Pg C, close to the 1997-2015 average (2.2 Pg C yr$^{-1}$) but ~25%
lower than 1997–98 — the difference mainly occurring in the southern tropics (0–23.5°S, Fig.
S10). In particular, carbon emissions from deforestation and peat fires were two times lower in
2015 (0.6 Pg C) compared with 1997 (1.2 Pg C) (GFED4s data). Emissions from these fire types
are more likely to be a net carbon source, because they cannot be compensated by vegetation
regrowth within a short time. Fire emission data thus suggest a smaller contribution from fires to
AGR in 2015 than 1997–98.

There has been a long debate on whether tropical vegetation shows enhanced greenness as
indicated by vegetation indices (i.e., NDVI and enhanced vegetation index or EVI) during dry
seasons or drought periods in tropical forest (Bi et al., 2015; Huete et al., 2006; Morton et al.,
2014; Saleska et al., 2007; Samanta et al., 2010; Xu et al., 2011), and whether there is an
accompanying decrease in long-term vegetation productivity associated with droughts (Medlyn,
2011; Samanta et al., 2011; Zhao and Running, 2010). Some studies show enhanced green-up in
Amazonian forest during dry seasons mainly due to the release of radiation control on vegetation
activities (Bi et al., 2015; Huete et al., 2006; Myneni et al., 2007), while Morton et al. (2014)
argued that if errors of satellite observation angle are corrected, no increase in EVI can be
observed during dry seasons. Saleska et al. (2007) observed greener response of Amazonian
forest during a severe drought event, whereas Samanta et al. (2010) argued such observed green-
up is an artefact of atmosphere-corrupted data and the properly processed satellite observations
reveal browner Amazonian forests during the severe drought event. Subsequent studies by Bi et
al. (2016) and Xu et al. (2011) confirm the satellite-observed negative impacts of the drought
events.

While long-term forest plot data demonstrated consistent negative effect of droughts on tropical
carbon uptake mainly through enhanced tree mortality (Lewis et al., 2011; Phillips et al., 2009),
short-time site observations failed to reveal immediate reduction in forest net primary
productivity (Doughty et al., 2015) during drought, or reported even increased gross
photosynthesis or photosynthetic capacity when entering dry season (Huete et al., 2006; Wu et
al., 2016). Further, a large mortality event for trees will cause a legacy source over several years
rather than a rapid release of $CO_2$ to the atmosphere during the year when trees died. Therefore,
it remains challenging to reconcile immediate carbon uptake reduction on the occurrence of
drought in tropical ecosystems as diagnosed from atmospheric inversions (Gatti et al., 2014) and
the aforementioned findings from forest plot data. In dynamic global vegetation models
(DGVMs), the interannual variations of simulated land carbon sink are dominated by those in net
primary production (Wang et al., 2016), which contradicts the site-level observations by
Doughty et al. (2015).

Both Wang et al. (2013) and Wang et al. (2014) found a higher correlation coefficient between
interannual variability in tropical land carbon fluxes (as inferred from interannual variations in
AGR) with that in temperature than in precipitation, which is confirmed by our analysis of
inversion-based tropical land flux anomalies with climate variations (Fig. 4). However, forest
plot observations point to the prevailing drought as the dominant factor reducing forest carbon
storage (Phillips et al., 2009). The need thus remains to reconcile the findings of temperature
dominance at large spatial scale and precipitation/moisture dominance at fine scale. Recently,
Jung et al. (2017) suggested that the dominant role of soil moisture over land carbon flux
anomalies shifts to temperature when the scale of spatial aggregation increases, due to the
compensatory water effects in the process of spatial upscaling. We also find that for all seasons
except Q3, inversion-based land carbon uptake anomalies in the tropics and southern extratropics
are positively correlated with soil water content (data not shown), with 2015 having an extreme
low soil water content anomaly in Q4, echoing the extreme high temperature anomaly shown in
Fig. S4l. This might indicate that temperature impacts the land carbon uptake mainly by
increasing evaporative demand and decreasing soil water content.

**4.3 Data uncertainties and perspective**
On the global and hemispheric scales, the inversion-derived land- and ocean-atmosphere fluxes
are well constrained by the observed atmospheric $CO_2$ growth rates at measurement sites.
However, because the observational network is heterogeneous and sites are sparsely distributed
(Supplementary Material Fig. S11), land $CO_2$ fluxes cannot be resolved precisely over each grid
cell (Kaminski et al., 2001) and some regions are better constrained than others. This could
hinder the precise pixel-scale matching between gridded $CO_2$ flux maps and climate states or the
occurrence of climate extremes to investigate how climate extremes have affected carbon fluxes.
Although we have identified that carbon uptake transitions for some regions and seasons might
be related to certain climate extremes (e.g., the role of precipitation in TeNH of Q4 shown in Fig.
S5h), in general exact attribution of these transitions into different climate drivers could be
elusive. Further, a few other uncertainties matter for the specific objective of this study. First, the
atmospheric network increased over time, so that the inversions have a better ability to detect and
quantify a sharp transition in $CO_2$ fluxes occurring in the last than in the first decade of the
period analysed. This might hide the detection of other more extreme end-of-year carbon
transitions during early years of our target period (1981-2015). Second, because measurements
for early 2016 are not used in the CAMS inversion and are not completely available in the Jena
inversion, the constraining of the last season in 2015 is weaker than for the other three seasons.
This could influence estimating the exact magnitude of the extreme Q4 negative anomaly in land
carbon uptake in 2015. Third, the sparse network of sites in boreal Eurasia and the tropics might
diminish the ability of inversion systems to robustly allocation carbon fluxes spatially, which
could yield high uncertainty in the carbon fluxes diagnosed for these regions (van der Laan-
Luijkx et al., 2015; Stephens et al., 2007).

Despite these uncertainties, the strong transition of $CO_2$ fluxes from Q3 to Q4 is the largest ever
found in the inversion records analysed here. Although 2015 shows extreme greening in the
Northern Hemisphere, this strong greenness has been only translated into a moderate annual
carbon sink anomaly in 2015, because vegetation greenness and land uptake anomalies are
largely decoupled outside the growing season. The strong transition to a carbon source in TeNH
in Q4 is consistent with the high precipitation that might have led to a large increase in
respiration loss.

In the tropics, the transition to a strong source in TroSH in Q4 is congruent with the expected
response of ecosystems to the peak of an El Niño event. However, given the ambiguous findings
regarding changes in vegetation greenness during dry seasons or drought periods by previous
studies (Saleska et al., 2007; Xu et al., 2011), and the uncertain roles of climate variations in
driving the regional land carbon balance, more work is needed to reveal how these processes
have evolved during opposing ENSO phases (i.e., the cold phase of La Niña versus the warm
phase of El Niño). Furthermore, large-scale spatial observation-based analysis is hampered by
the scarcity of sites measuring atmospheric concentrations or land-atmosphere fluxes with the
eddy covariance method (Tramontana et al., 2016). For the boreal and temperate Northern
Hemisphere, further investigation is still needed to verify whether a coupling between strong
spring/summer uptake and autumn release is something intrinsic to natural ecosystems, or if
strong transitions to autumn release are triggered by some particular extreme climate shifts. More
detailed mechanisms can be explored by using long-term simultaneous observations of
vegetation greenness and eddy covariance measurements of land-atmosphere fluxes combined
with dynamic vegetation models. Here, our results point to the need to better understand the
drivers of carbon dynamics at seasonal, or even shorter time scales at the regional to global level,
especially the link between such dynamics and climate extremes. Such understanding would help
us make better predictions of the response of the carbon cycle to multiple long-term drivers such
as atmospheric $CO_2$ growth and climate change.

**5 Conclusions**
We investigated the links among vegetation greenness, interannual land carbon flux variations
and climate variations for 1981–2015 using inversion-based land carbon flux data sets.
Consistent positive correlations between satellite-derived vegetation greenness and land carbon
uptakes are found for the Northern Hemisphere during the growing season, but outside the
growing season, vegetation greenness and land carbon uptake are largely decoupled. Carbon
uptake in the boreal Northern Hemisphere (>45°N) is more consistently associated with
temperature than precipitation, although such a pattern is less evident for the temperate Northern
Hemisphere (23.5–45°N). Consistent with previous studies, we found a strong negative impact
by temperature in the land carbon uptakes in the tropics and Southern Hemisphere, probably
driven by the role of temperature in soil water content that tends to induce drought conditions.

We put an emphasis on the seasonal dynamics of land carbon uptake in 2015. We found that
northern lands started with a higher-than-normal sink for the northern growing season, consistent
with enhanced vegetation greenness partly owing to elevated warming. However, this enhanced
sink was in part balanced by the carbon release in the autumn and winter, associated with
extremely high precipitation in Q4 in the temperate Northern Hemisphere (23.5–45°N). Our
results emphasized the important role of the coupling between seasonal carbon dynamics in the
annual net carbon balance of the land ecosystem. More research is needed on whether such a
coupling, between enhanced spring and summer sink and reduced autumn uptake, is something
intrinsic in northern ecosystems, and on the frequency and extent of its occurrence. The
dominance of temperature in the boreal Northern Hemisphere (>45°N) and soil moisture in the
temperate Northern Hemisphere (23–45°N) in their autumn carbon loss implies that future
autumn temperature and precipitation change could have important consequences for the annual
carbon balance of these regions. Hence, although continuing vegetation greening is projected
mainly thanks to $CO_2$ fertilization (Zhu et al., 2016), such greening might not translate into
enhanced land carbon uptake.

For the tropics and Southern Hemisphere, a strong transition was found towards a large carbon
source for the last quarter of 2015, concomitant with the peak of El Niño development. This
strong transition of terrestrial $CO_2$ fluxes in the last season is the largest in the inversion records
since 1981, even though annual fire emissions were ~25% lower than during El Niño of 1997–
98. However, site-scale studies on tropical forest growth so far focusing on drought impacts
cannot provide an adequate explanation of such strong transitions. It is unclear how the
individual fluxes (gross primary production, net primary production, heterotrophic respiration)
that make up the land sink have responded to drought conditions. Our results point to the
possibility that, with more frequent extreme El Niño events being projected in the future, such
strong seasonal transitions in land carbon uptake might become more frequent and they can have
substantial impact on the capacity of land ecosystems to sequester carbon.

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

**Acknowledgements**

C.Y. and P.C. acknowledge funding from the European Commission's 7th Framework Programme, under grant agreement number 603542 (LUC4C). The work of F.C. was funded by the Copernicus Atmosphere Monitoring Service, implemented by the European Centre for Medium-Range Weather Forecasts (ECMWF) on behalf of the European Commission. Taejin Park was supported by the NASA Earth and Space Science Fellowship Program (grant no. NNX16AO34H). We thank all the scientists involved in the surface and aircraft measurement of atmospheric $CO_2$ concentration and in archiving these data and making them available. We also thank Dr. Matthias Forkel and the anonymous reviewer for their comments that helped improve the quality of the manuscript. We thank Mr. John Gash for improving the English of our manuscript.

**Author contributions**

P.C., F.C., C.Y. and A.B. conceived the study. C.Y. performed the analysis and made the first draft. F.C. and C.R. provided the inversion data. T. P. provided the NDVI data. All authors contributed to the interpretation of the results and writing of the paper.

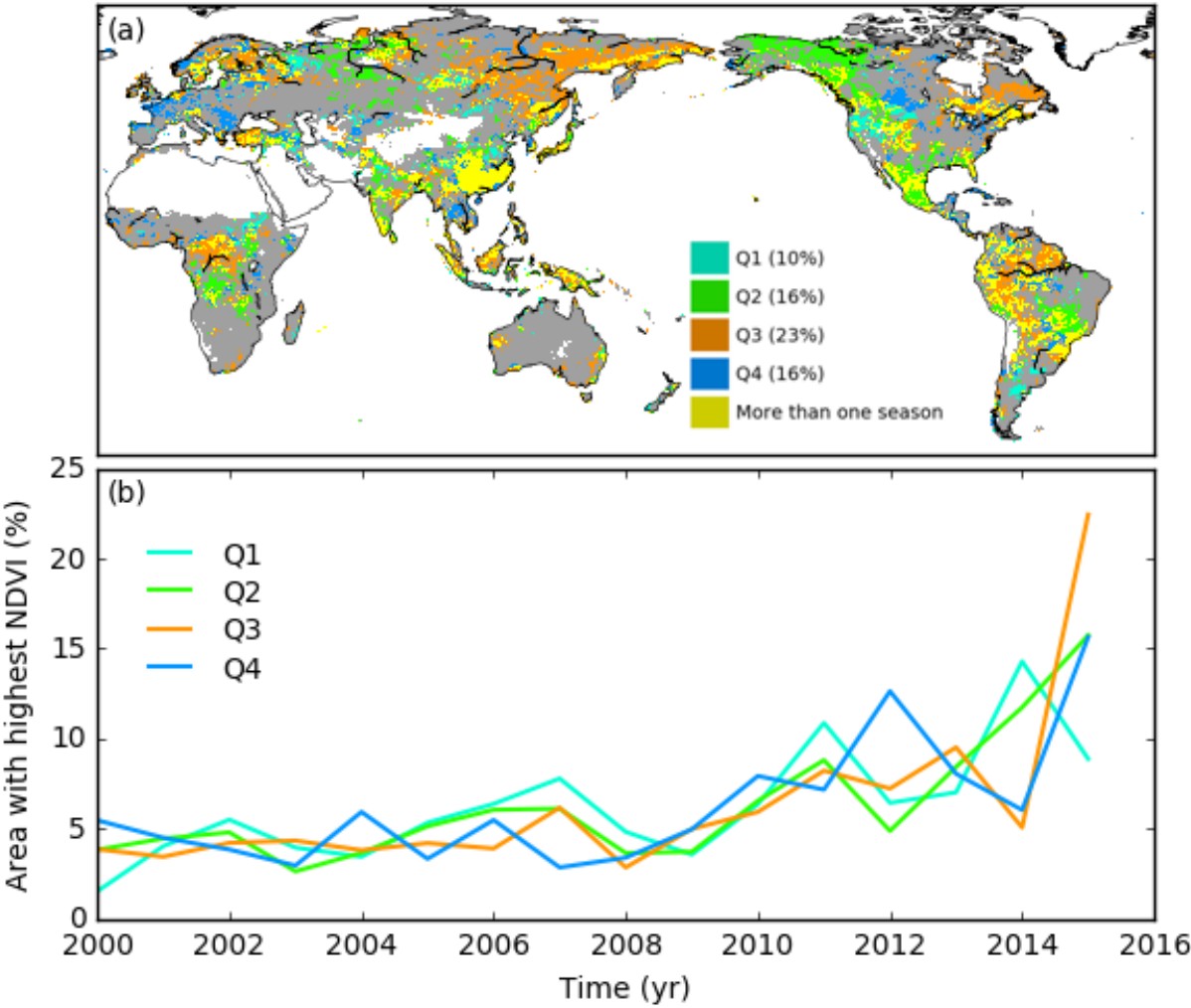


**Figure 1** Year 2015 as the greenest year over the period 2000-2015. (a) Distribution of seasons
for which 2015 NDVI ranks the highest during the period 2000-2015. Yellow-coloured pixels
indicate grid cells where 2015 NDVI ranks highest for more than one season. For each season,
the fraction of global vegetated land area for which 2015 NDVI ranks highest is shown in the
inset colour scale. (b) Temporal evolution of the percentage of vegetated land with highest NDVI
over 2000-2015 for each season and different years. The sum total of vertical-axis values for
each season over all years is 100%. Q1 = January–March; Q2 = April–June; Q3 = July–
September; Q4 = October–December.

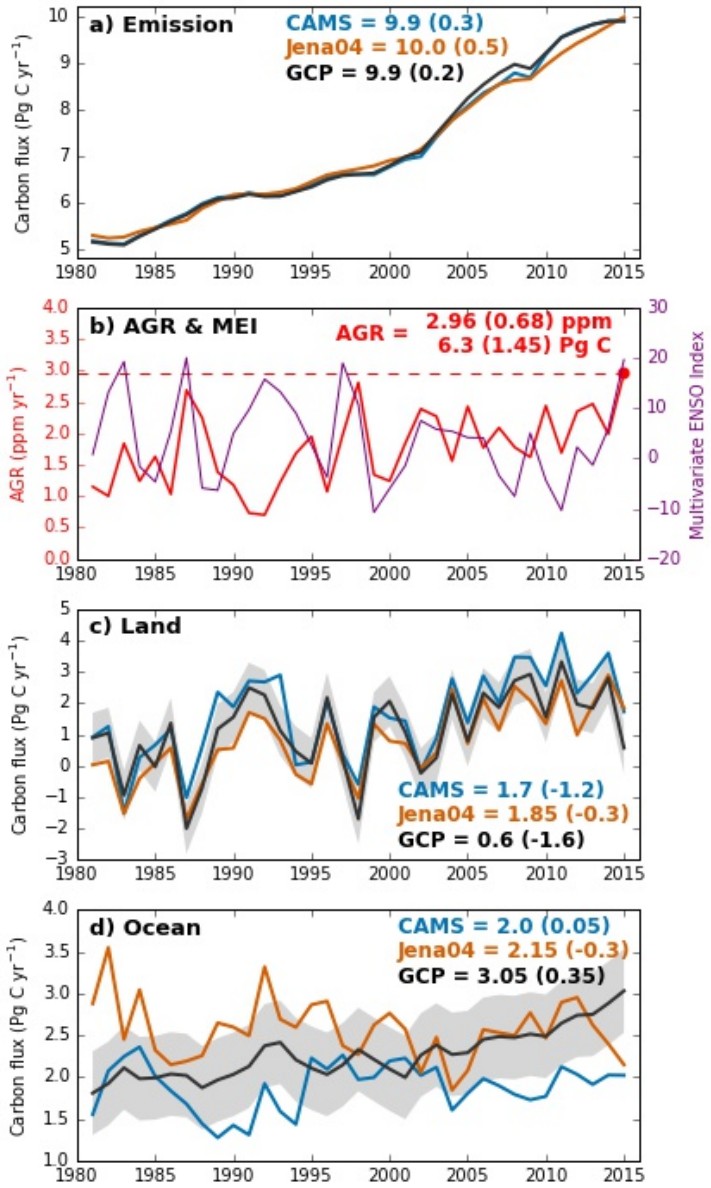


**Figure 2** Global carbon fluxes and atmospheric $CO_2$ growth rates for 1981–2015. (a) Carbon
emissions from fossil fuel and industry used in the CAMS (blue) and Jena04 (orange) inversions,
(b) annual atmospheric $CO_2$ growth rate (AGR, in red) from NOAA/ESRL linked to Multivariate
ENSO Index (in purple), and (c) land and (d) ocean carbon sinks for 1981-2015. Emissions, land
and ocean carbon sinks from the Global Carbon Project (GCP, in black) are also shown for
comparison. In subplots (c) and (d), a carbon flux of 0.45 Pg C yr$^{-1}$ was used to correct
inversion-derived land and ocean sinks to account for pre-industrial land-to-ocean carbon flux as
in Le Quéré et al. (2016). All numbers indicate values in 2015 (Pg C yr$^{-1}$, rounded to ±0.05 Pg C
yr$^{-1}$), with those in brackets showing linearly de-trended anomalies for the same year.

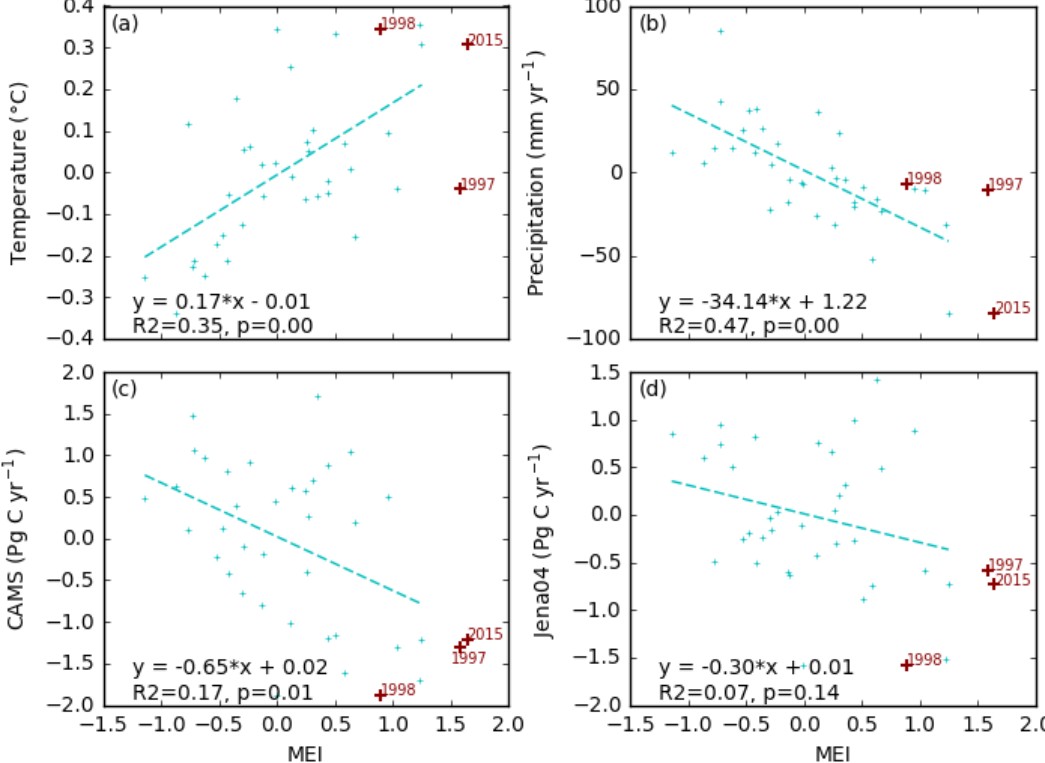

**Figure 3** Relationships between anomalies of (a) land air temperature, (b) land precipitation, (c)
land carbon fluxes by the CAMS inversion, (d) land carbon fluxes by the Jena04 inversion, and
the Multivariate ENSO Index (MEI). All variables are linearly de-trended over 1981–2015.

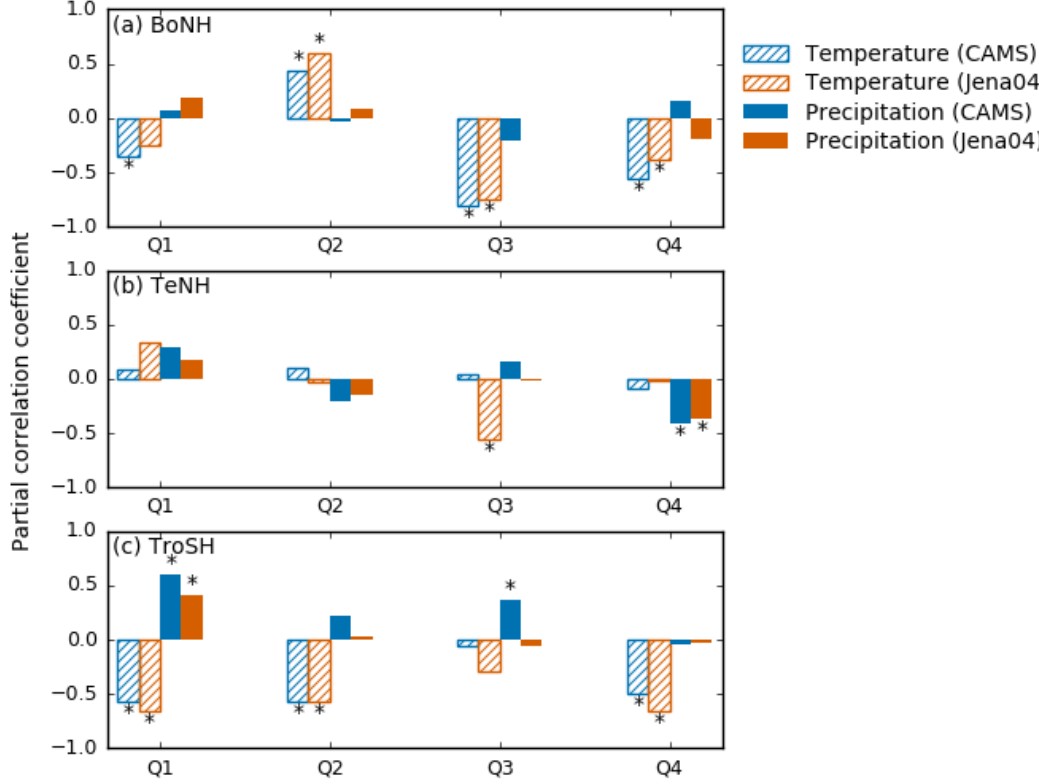

**Figure 4** Partial correlation coefficients of de-trended annual anomalies of land carbon fluxes by CAMS and Jena04 inversions against the anomalies in temperature and precipitation of different seasons. n = 34. An asterisk indicates significant correlation (p<0.05).

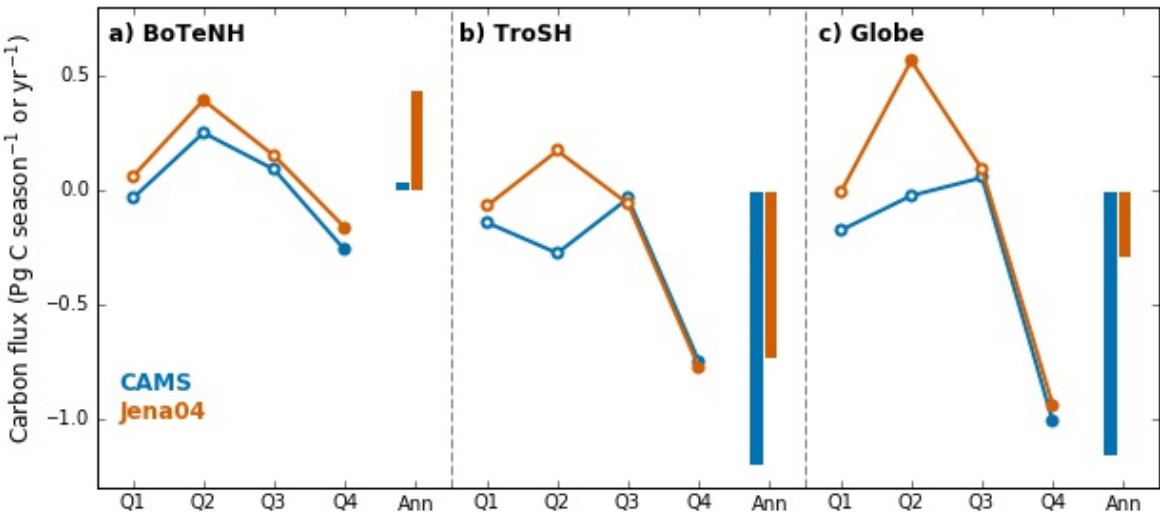


**Figure 5** Seasonal land carbon uptake anomalies in 2015. Data are linearly de-trended over
1981-2015 for different seasons in 2015, by CAMS (blue) and Jena04 (orange) inversion data.
Open or solid dots indicate seasonal values (Pg C season$^{-1}$) and vertical bars indicate annual sum
(Pg C yr$^{-1}$). Data are shown for: (a) boreal and temperate Northern Hemisphere (BoTeNH, >
23.5°N), (b) tropics and extratropical Southern Hemisphere (TroSH, < 23.5°N) and (c) the whole
globe. Solid dots indicate seasonal land carbon uptake anomalies below 10th or above 90th
percentiles over 1981-2015.

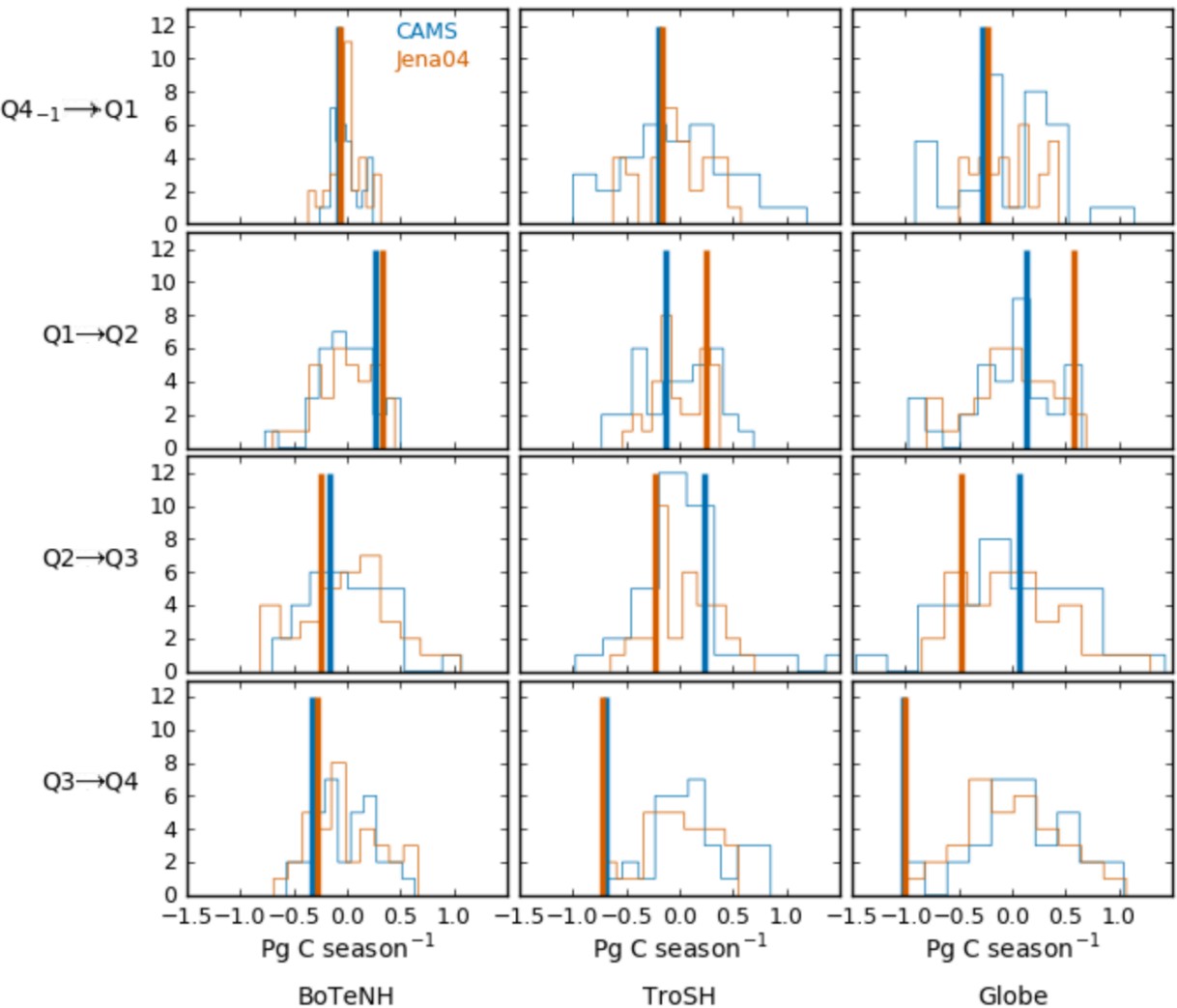

**Figure 6** Extremeness of transitions in seasonal land carbon uptake anomaly in 2015. Lines of histograms for seasonal land carbon uptake transitions over 1981-2015 are shown for boreal and temperate Northern Hemisphere (BoTeNH, latitude > 23.5°N), tropics and extratropical Southern Hemisphere (TroSH, latitude < 23.5°N) and the whole globe. Transition between two consecutive seasons is defined as the linearly de-trended land carbon uptake anomaly in a given season minus that in the former one. Horizontal-axis shows the seasonal transitions in land carbon uptake anomalies (Pg C season$^{-1}$). Vertical orange solid lines indicate values for 2015.