# Peer review of "variations: a focus on the year 2015"

_Atmospheric Chemistry and Physics, 2016_

## Referee Comment (RC1) · Anonymous Referee #1 · 19 Feb 2017

The article "Abrupt seasonal transitions in land carbon uptake in 2015" by C. Yue and coauthors presents a detailed analysis of anomalies in carbon sinks and sources, climate and vegetation greenness during recent decades with an emphasis on the year 2015. Understanding the carbon cycle and its interaction with climate change is a highly relevant research topic, and the authors refer to state-of-the-art literature and datasets. The authors combine a number of observational datasets and model results, and my impression is that their methods and results are sound. The description of the work steps is clear and the data sources are well documented. In this regard, the article is good at what it does.

My major concern however is that it remains unclear what the authors are trying to achieve with this article. I would guess that the results might tell us something about how climate affects vegetation and the carbon cycle. What do the results imply about

the relevant processes, about past climates and potential future developments, or about our potential to model these processes? The authors address such questions only briefly in the last paragraph of Sect. 4 and in the very short Sect. 5, stating that they go beyond the scope of the article.

I also wonder why the authors focus so much on the year 2015. What is so special about this year (apart from being relatively recent) that would justify this focus, and what can we learn from this case study that is valid in a greater context? If there is something I am overlooking, I suggest that the authors reframe their article to bring out their message more explicitly, and that they stress what the progress is compared to previous articles. I believe that this would improve the impact of their article. For example, the authors could systematically relate anomalies in climate, carbon fluxes and NDVI using the whole record, and not only focus on 2015. They should also consider to include the year 2016 (if possible) to capture the full recent El Nino event. It appears a bit arbitrary that they pick the year 2015 and one other previous El Nino event for their analysis, using the rest of their data only to calculate linear trends. A more comprehensive statistical analysis of the available data might allow more general conclusions without the need of running climate models.

So far, the main selling points of the paper seem to be

(i) the (arguably) counterintuitive combination of high NDVI and negative carbon uptake anomaly

(ii) large anomaly of the year 2015.

Regarding (i), I find it little surprising that greening and carbon loss (or a reduced carbon sink) can go together since both anomalies can be dominated by different locations and different seasons, and because they are not linearly related given the complex ecological processes involved. The authors point this out themselves, hence invoking a "paradox" seems a little exaggerated to my taste. (But I would be curious if this anti-correlation is a temporal feature or a robust trend that can be expected to continue in

the future; something the authors might choose to give more attention to.) Regarding (ii), I find it misleading to speak of an "abrupt transition" (title, abstract and line 263+). This term gives the impression of a singular event with long-lasting consequences, like a forced non-reversible switch to another state or regime. However, the phenomenon discussed in the paper appears to be an anomaly that is the realisation of natural variability, hence an extreme but temporal event. This comes on top of a gradual trend to larger growth rates, so the year 2015 will most likely not be unique. In fact, the atmospheric growth rate of $CO_2$ in 2016 was even higher than in 2015. I therefore wonder whether the term "abrupt transition" is useful here, and would suggest a more suitable term, e.g. the land carbon uptake anomaly in 2015. I therefore also strongly suggest a change in the paper's title.

Minor comments

- line 85-86: "We used ... includes"

- line 89: What is a validity period, and in what sense are the other years are not valid?

- line 108: Why do the authors pick MAI to characterise the ENSO state?

- line 141-148. At the first reading I did not understand the role of the "historical trend" for the growth rate in a given year. I understand now that specific anomalies in 2015 are later related to climate anomalies, with anomalies being defined as residuals after removing a linear trend. The reasoning behind this could be explained more explicitly here.

- line 201: both instead of bother

- line 136: It would help non-experts to briefly explain how the sources and sinks are quantified in the GCP. How independent is this dataset from the inversion calculations?

- Sect. 2.2.1: It would help me to already see time series and a map as a visualisation of the rank analysis. I understand the structure of the paper and find it reasonable, but it could make sense to merge the data analysis section 2.2 with Results Sect.

3. Otherwise, one has to read the methods section without visualisation, and later remember each methodological detail when the results are shown. This is a matter of taste and I leave it to the authors to reconsider the structure.

- line 226: "the seemingly paradox" is grammatically wrong.

- line 350: data suggests (not suggest)

- Supplementary Material: I suggest to put captions underneath (not above) the figures and increase the space between the figures. There is too much space between the caption of Fig. 2 and Fig. 2. These things make it difficult to identify the right caption for each figure.

―――――――――――――――――――――

---

## Referee Comment (RC2) · M. Forkel (Referee) · 8 Mar 2017

The comment was uploaded in the form of a supplement:
http://www.atmos-chem-phys-discuss.net/acp-2016-1167/acp-2016-1167-RC2-supplement.pdf

---

## Author Comment (AC1) · 14 Jun 2017

We thank the reviewer for the constructive comments. Please find in the attached file our responses.
* * *
[Figure]
The article "Abrupt seasonal transitions in land carbon uptake in 2015" by C. Yue and coauthors presents a detailed analysis of anomalies in carbon sinks and sources, climate and vegetation greenness during recent decades with an emphasis on the year 2015. Understanding the carbon cycle and its interaction with climate change is a highly relevant research topic, and the authors refer to state-of-the-art literature and datasets. The authors combine a number of observational datasets and model results, and my impression is that their methods and results are sound. The description of the work steps is clear and the data sources are well documented. In this regard, the article is good at what it does.

My major concern however is that it remains unclear what the authors are trying to achieve with this article. I would guess that the results might tell us something about how climate affects vegetation and the carbon cycle. What do the results imply about the relevant processes, about past climates and potential future developments, or about our potential to model these processes? The authors address such questions only briefly in the last paragraph of Sect. 4 and in the very short Sect. 5, stating that they go beyond the scope of the article.

[Response] We thank the general positive comments by the reviewer. We were originally aiming for two purposes in this article: (a) to diagnose the anomaly of large scale $CO_2$ fluxes for 2015 given the specific nature of that year, as a case study (high $CO_2$ growth rate, anomalously strong vegetation greenness and the historically highest annual temperature), using atmospheric inversion data, and (b) to diagnose whether abrupt transitions have occurred in terrestrial carbon uptake in 2015, and briefly infer the reasons for such transitions.

We agree with reviewer that the exploration of the general links among vegetation greenness, land carbon uptake dynamics and climate variations is necessary in order to put the 2015 case into a more general picture, to infer general patterns of land carbon dynamics that could be useful for future prediction of land carbon dynamics. We also add this point as one of the research aims of our paper. According changes are made in revised abstract, and the 3$^{rd}$ paragraph of the revised Introduction section.

We have extensively revised the manuscript to incorporate correlations of land carbon uptake anomalies with vegetation greenness anomalies and climate anomalies related with ENSO dynamics. Two new figures (Fig. 3, Fig. 4) are added in the main text, and three new figures (Fig. S4, S5, S7) are added in the Supplementary Material. Results and discussion sections are substantially expanded to include more discussions on the mechanisms underlying land carbon dynamics, and the relevance of this study.

I also wonder why the authors focus so much on the year 2015. What is so special about this year (apart from being relatively recent) that would justify this focus, and what can we learn from this case study that is valid in a greater context? If there is something I am overlooking, I suggest that the authors reframe their article to bring out their message more explicitly, and that they stress what the progress is compared to previous articles. I believe that this would improve the impact of their article. For example, the authors could systematically relate anomalies in climate, carbon fluxes and NDVI using the whole record, and not only focus on 2015. They should also consider to include the year 2016 (if possible) to capture the full recent El Nino event. It appears a bit arbitrary that they pick the year 2015 and one other previous El Nino event for their analysis, using the rest of their data only to calculate linear trends. A more comprehensive statistical analysis of the available data might allow more general conclusions without the need of running

**Fig. 1.**

---

## Author Comment (AC2) · 14 Jun 2017

We thank Matthias Forkel very much for the constructive comments. Please find attached our responses.

[Figure]
* * *
**Review of "Abrupt seasonal transitions in land carbon uptake in 2015" by Chao Yue et al.**

Matthias Forkel, 2017-03-08

**1. Does the paper address relevant scientific questions within the scope of ACP?**

The article by C. Yue et al. addresses annual and seasonal variabilities in global land carbon uptake and the relations with climate and vegetation. This paper is within the scope of ACP.

**2. Does the paper present novel concepts, ideas, tools, or data?**

The paper is based on well established datasets and methods to generate such data ($CO_2$ measurements, NDVI data, atmospheric inversion). The title and the abstract of the paper mainly highlights one finding of the study about "abrupt seasonal transitions in land carbon uptake". This finding is not really new (except the focus on 2015) but the results of the study are a good opportunity to remind the land carbon cycle community about such mechanisms and to point to the year 2015 as a remarkable example of such seasonal transitions.

**3. Are substantial conclusions reached?**

The entire study is focussed on anomalies of the land carbon uptake in the year 2015 relative to the period 1981 to 2015. Consequently, the conclusions are very specific for climate/carbon cycle mechanism in this year. To make this paper more interesting for the land carbon cycle community and to reach more substantial and less specific conclusions, I would recommend to perform similar analyses also for other years and to finally draw conclusions about general mechanisms in comparison to specificities in single years. In this point, I completely agree with Anonymous Referee #1.

[Response] We examined extensively the relationship between anomalies in land carbon uptake, NDVI and climate variations. These new analyses are incorporated in the substantially revised results and discussion section.

**4. Are the scientific methods and assumptions valid and clearly outlined?**

Overall, yes. For some datasets, I would expect scientific references additionally to the URLs from which the data was obtained (especially in Sections 2.2.2 and 2.2.3). The only exception is the analysis of NDVI data (Section 2.2.1): For example, the authors calculated "seasonal mean standardized NDVI". Although I have some experience with NDVI data (Forkel et al., 2013), I cannot imagine what this term means. How were NDVI values standardized? Why? Furthermore, mean NDVI values of winter seasons in northern

[Figure]

**Fig. 1.**

---

## Author Comment (AC3) · 14 Jun 2017

The article "Abrupt seasonal transitions in land carbon uptake in 2015" by C. Yue and coauthors presents a detailed analysis of anomalies in carbon sinks and sources, climate and vegetation greenness during recent decades with an emphasis on the year 2015. Understanding the carbon cycle and its interaction with climate change is a highly relevant research topic, and the authors refer to state-of-the-art literature and datasets. The authors combine a number of observational datasets and model results, and my impression is that their methods and results are sound. The description of the work steps is clear and the data sources are well documented. In this regard, the article is good at what it does.

My major concern however is that it remains unclear what the authors are trying to achieve with this article. I would guess that the results might tell us something about how climate affects vegetation and the carbon cycle. What do the results imply about the relevant processes, about past climates and potential future developments, or about our potential to model these processes? The authors address such questions only briefly in the last paragraph of Sect. 4 and in the very short Sect. 5, stating that they go beyond the scope of the article.

[Response] We thank the general positive comments by the reviewer. We were originally aiming for two purposes in this article: (a) to diagnose the anomaly of large scale $CO_2$ fluxes for 2015 given the specific nature of that year, as a case study (high $CO_2$ growth rate, anomalously strong vegetation greenness and the historically highest annual temperature), using atmospheric inversion data, and (b) to diagnose whether abrupt transitions have occurred in terrestrial carbon uptake in 2015, and briefly infer the reasons for such transitions.

We agree with reviewer that the exploration of the general links among vegetation greenness, land carbon uptake dynamics and climate variations is necessary in order to put the 2015 case into a more general picture, to infer general patterns of land carbon dynamics that could be useful for future prediction of land carbon dynamics. We also add this point as one of the research aims of our paper. According changes are made in revised abstract, and the 3$^{rd}$ paragraph of the revised Introduction section.

We have extensively revised the manuscript to incorporate correlations of land carbon uptake anomalies with vegetation greenness anomalies and climate anomalies related with ENSO dynamics. Two new figures (Fig. 3, Fig. 4) are added in the main text, and three new figures (Fig. S4, S5, S7) are added in the Supplementary Material. Results and discussion sections are substantially expanded to include more discussions on the mechanisms underlying land carbon dynamics, and the relevance of this study.

I also wonder why the authors focus so much on the year 2015. What is so special about this year (apart from being relatively recent) that would justify this focus, and what can we learn from this case study that is valid in a greater context? If there is something I am overlooking, I suggest that the authors reframe their article to bring out their message more explicitly, and that they stress what the progress is compared to previous articles. I believe that this would improve the impact of their article. For example, the authors could systematically relate anomalies in climate, carbon fluxes and NDVI using the whole record, and not only focus on 2015. They should also consider to include the year 2016 (if possible) to capture the full recent El Nino event. It appears a bit arbitrary that they pick the year 2015 and one other previous El Nino event for their analysis, using the rest of their data only to calculate linear trends. A more comprehensive statistical analysis of the available data might allow more general conclusions without the need of running

climate models.

So far, the main selling points of the paper seem to be

(i) the (arguably) counterintuitive combination of high NDVI and negative carbon uptake anomaly (ii) large anomaly of the year 2015.

Regarding (i), I find it little surprising that greening and carbon loss (or a reduced carbon sink) can go together since both anomalies can be dominated by different locations and different seasons, and because they are not linearly related given the complex ecological processes involved. The authors point this out themselves, hence invoking a "paradox" seems a little exaggerated to my taste. (But I would be curious if this anti- correlation is a temporal feature or a robust trend that can be expected to continue in the future; something the authors might choose to give more attention to.) Regarding (ii), I find it misleading to speak of an "abrupt transition" (title, abstract and line 263+). This term gives the impression of a singular event with long-lasting consequences, like a forced non-reversible switch to another state or regime. However, the phenomenon discussed in the paper appears to be an anomaly that is the realisation of natural variability, hence an extreme but temporal event. This comes on top of a gradual trend to larger growth rates, so the year 2015 will most likely not be unique. In fact, the atmospheric growth rate of $CO_2$ in 2016 was even higher than in 2015. I therefore wonder whether the term "abrupt transition" is useful here, and would suggest a more suitable term, e.g. the land carbon uptake anomaly in 2015. I therefore also strongly suggest a change in the paper's title.

[Response] (1) We now use the full 1981–2015 data and performed statistical analysis of vegetation greenness, land carbon uptake and climate anomalies for different regions and seasons. These results are incorporated in the revised manuscript in both result and discussion sections, with findings from previous studies being extensively referred to and discussed as well. (2) We maintain the "paradox" expression because we think it is adequate to describe the year 2015, which comes with extreme greenness and an only moderate land carbon sink. Higher greenness is sometimes simply assumed to be associated with higher sink, but this is not necessarily true, as is also pointed out by the reviewer. We now examine in more detail in the revised the relationship between land carbon uptake and vegetation greenness for different seasons and regions. (3) As explained in the response to the previous comment, now the manuscript is restructured around two research aims: to examine general relationships among vegetation greenness, land carbon uptake and climate variations, and to examine the 2015 as a special case on how land carbon dynamics have responded to a combination of extreme greenness and ENSO climate variations. This is made clear in the revised manuscript. (4) We mostly drop the word 'abrupt' given its potential confusion in an ecological context and, instead, the word "strong" is used. (5) We change the title to reflect the revision in the manuscript content to "Vegetation greenness and land carbon flux anomaly associated with climate variations with a special focus on 2015". (6) We did not include the year 2016 into the current analysis because the inversion data are not available yet. But we believe focusing on the year 2015 could already generate meaningful conclusions from our manuscript.

Minor comments

- line 85-86: "We used ... includes"

[Response] we changed 'includes' to 'including'.

- line 89: What is a validity period, and in what sense are the other years are not valid?

[Response] Site observations used in the Jena CarboScope inversion are coherent over time within the so-called "validity period", but are not outside the validity period. More specifically, the validity period is

defined by the one using a consistent number of sites, i.e., all sites that have observations over such a period. Outside the validity period, site numbers changed depending on their availability or operation time. It is optimal to examine the temporal trend within the validity period, but this does not mean the data outside this period are invalid and should not be used. In fact, the same situation also happens in CAMS inversion, which considers a variable number of sites during the full study period. Because our analysis has to reconcile the need of a large site number in 2015 and long historical period for a robust anomaly estimate, using the s04_v3.8 run outside its validity period is therefore a compromise. Besides the responses here, we made the according changes in Section 2.1.1.

- line 108: Why do the authors pick MAI to characterise the ENSO state?

[Response] We believe the reviewer means MEI rather than MAI. MEI (Multivariate ENSO Index) is the first unrotated principal component of six variables over the tropical Pacific that are closely linked with ENSO. Among the six variables sea-level pressure, sea surface temperature and surface air temperature are included. MEI has been widely used in literature as an indicator for the ENSO state, for instance, Nemani et al., 2003; Wang et al., 2013; van der Werf et al., 2008. The MEI should, therefore, summarize not only the ocean component of ENSO (El-Niño), but also the atmospheric component (the Southern Oscillation).

As a complement to MEI, we also used the Oceanic Niño Index (ONI, http://www.cpc.ncep.noaa.gov/products/analysis_monitoring/ensostuff/ONI_change.shtml) when comparing the evolution of El Niño events of 1997 and 2015 in Supplementary Figure S10. The ONI tracks the running 3-month average sea surface temperatures in the east-central tropical Pacific between 120°-170°W (the Niño 3.4 region). Supplementary Figure S10 shows very similar temporal patterns of MEI and ONI during El Niño evolution especially when El Niño reached its peak, indicating the suitability of MEI being used in ENSO-related analysis.

The pieces of information described above are also included in the revised manuscript in appropriate sections.

- line 141-148. At the first reading I did not understand the role of the "historical trend" for the growth rate in a given year. I understand now that specific anomalies in 2015 are later related to climate anomalies, with anomalies being defined as residuals after removing a linear trend. The reasoning behind this could be explained more explicitly here.

[Response] We added the following text in this paragraph to make it more explicit and hope it will help clarify better: "The record-breaking AGR in 2015 thus must be put into an historical perspective to reconcile evidence for extreme greening and the highest atmospheric $CO_2$ growth rate. For example, if 2015 comes up with a large increase in carbon emissions accompanied by droughts (browning) in the northern hemisphere and the tropics, then the highest AGR might not be regarded as a big surprise. Therefore, to understand the contributing factors for the highest AGR in 2015, it must be separated into a long-term trend and interannual anomalies."

- line 201: both instead of bother

[Response] This has been corrected.

- line 136: It would help non-experts to briefly explain how the sources and sinks are quantified in the GCP. How independent is this dataset from the inversion calculations?

[Response] Estimates of land and ocean carbon uptakes are largely independent from the two inversions

used in this study. We have inserted the following text in this paragraph to clarify this: "Estimates of ocean carbon uptake in GCP are based on observation-based mean $CO_2$ sink estimate for the 1990s and variability in the ocean $CO_2$ sink for 1959–2015 from global ocean biogeochemistry models. Estimates of land carbon uptake in GCP are calculated as the difference between anthropogenic emissions, atmospheric $CO_2$ growth and ocean sink. The estimates of land and ocean carbon uptake in GCP are largely independent from the two inversions used here, except that the $CO_2$ records from atmospheric stations which are used in inversions are also used in GCP to derive global AGR."

- Sect. 2.2.1: It would help me to already see time series and a map as a visualisation of the rank analysis. I understand the structure of the paper and find it reasonable, but it could make sense to merge the data analysis section 2.2 with Results Sect. 3. Otherwise, one has to read the methods section without visualisation, and later remember each methodological detail when the results are shown. This is a matter of taste and I leave it to the authors to reconsider the structure.

[Response] Relevant figures (Fig. 1, Supplementary Fig. S1, Supplementary Fig. S3) are now cited in this section in the revised manuscript to help readers understand better the methods. But we maintained the methods and result as two separate sections mainly for the clarity of the structure.

- line 226: "the seemingly paradox" is grammatically wrong.

[Response] "seemingly" is changed to "seeming".

- line 350: data suggests (not suggest)

[Response] Corrected.

- Supplementary Material: I suggest to put captions underneath (not above) the figures and increase the space between the figures. There is too much space between the caption of Fig. 2 and Fig. 2. These things make it difficult to identify the right caption for each figure.

[Response] Figure captions are now put below figures.

---

## Author Comment (AC4) · 14 Jun 2017

**Review of "Abrupt seasonal transitions in land carbon uptake in 2015" by Chao Yue et al.**

Matthias Forkel, 2017-03-08

**1. Does the paper address relevant scientific questions within the scope of ACP?**

The article by C. Yue et al. addresses annual and seasonal variabilities in global land carbon uptake and the relations with climate and vegetation. This paper is within the scope of ACP.

**2. Does the paper present novel concepts, ideas, tools, or data?**

The paper is based on well established datasets and methods to generate such data ($CO_2$ measurements, NDVI data, atmospheric inversion). The title and the abstract of the paper mainly highlights one finding of the study about "abrupt seasonal transitions in land carbon uptake". This finding is not really new (except the focus on 2015) but the results of the study are a good opportunity to remind the land carbon cycle community about such mechanisms and to point to the year 2015 as a remarkable example of such seasonal transitions.

**3. Are substantial conclusions reached?**

The entire study is focussed on anomalies of the land carbon uptake in the year 2015 relative to the period 1981 to 2015. Consequently, the conclusions are very specific for climate/carbon cycle mechanism in this year. To make this paper more interesting for the land carbon cycle community and to reach more substantial and less specific conclusions, I would recommend to perform similar analyses also for other years and to finally draw conclusions about general mechanisms in comparison to specificities in single years. In this point, I completely agree with Anonymous Referee #1.

[Response] We examined extensively the relationship between anomalies in land carbon uptake, NDVI and climate variations. These new analyses are incorporated in the substantially revised results and discussion section.

**4. Are the scientific methods and assumptions valid and clearly outlined?**

Overall, yes. For some datasets, I would expect scientific references additionally to the URLs from which the data was obtained (especially in Sections 2.2.2 and 2.2.3). The only exception is the analysis of NDVI data (Section 2.2.1): For example, the authors calculated "seasonal mean standardized NDVI". Although I have some experience with NDVI data (Forkel et al., 2013), I cannot imagine what this term means. How were NDVI values standardized? Why? Furthermore, mean NDVI values of winter seasons in northern

regions are not very useful to draw conclusions about vegetation productivity or land carbon uptake. As NDVI is a land surface property it is not only affected by vegetation but outside the peak of the growing season strongly by changes in snow cover and soil reflectance. Consequently, a certain ranking in a season espcially in northern regions might be due to the variability in snow cover but not in vegetation. The authors need to appropriate filter the NDVI time series to separate vegetation signals from other non-vegetation distortions (Hird and McDermid, 2009; Holben, 1986; Kandasamy et al., 2013). Furthermore, NDVI datasets from different sensors show large differences which are especially important for seasonal anomalies that are outside of the peak of the growing season (D'Odorico et al., 2014; Fensholt and Proud, 2012; Kern et al., 2016; Scheftic et al., 2014). Consequently, I'm wondering if the shown ranking of seasonal NDVI values (Fig. 1) is a robust result given the noise of NDVI data and the differences between datasets. This rises the question if 2015 is indeed the greenest year.

[Response] The scientific citations for MEI and NDVI are provided in additional to URL links. NDVI reflects in general vegetation green fraction, and is considered as a proxy of green leaf area (Gamon et al., 1995; Ide et al., 2010) Its temporal magnitudes have been used to infer changes in vegetation productivity (Myneni et al., 1997; Zhao and Running, 2010). In response to the reviewer's comments, we have updated our results by using a new NDVI data set that went through rigorous quality control, with the cloud- and snow-contaminated pixels being removed and gap-filled. Note that the original NDVI values, rather than standardized anomalies, are used. Seasonal NDVI values lower than 0.1 were further removed to make sure the used NDVI values reflect the dominance of vegetation information.

Because of filtering NDVI values by a minimum of 0.1, we are cautiously confident that the vegetation greenness reflects (at least partly) the vegetation information even in the first (Q1) and fourth (Q4) trimester of the year when snow is present in the northern hemisphere. In fact, October is frequently considered within the growing season and some evergreen coniferous forests show significant photosynthetic activities in March in regions of mild winter, e.g., Tanja et al., 2003). Here we show that, most of the grid cells where 2015 shows the highest NDVI for northern land (Fig. CS1, region with latitude > 30°N) are in fact dominated by Q2 (April–June) and Q3 (July–September), corresponding roughly to northern hemisphere growing season. The vegetated land area (i.e., with a seasonal NDVI value higher than 0.1 in either of the four seasons of 2015) with the highest NDVI rank in 2015 in the northern land accounts for 36% of all land area, in contrast with an expected mean of 6.25% if the land is equally green over all years of 2000–2015. This highlights again the extreme greening during the growing season in the northern hemisphere in 2015, as has been examined in more detail in Bastos et al. (2017).

Q1 and Q4 account for 34% of the land area where 2015 NDVI ranks the highest in northern land. These grid cells are either dominated by evergreen forests (central Canada, northwestern Europe), or by oceanic climate where evergreen forests prevail (eastern Canada and US, Europe) (Fig. CS2). As shown in Fig. CS3, the land north to 23.5°N contributes primarily to the overall highest annual NDVI in 2015, whereas in tropics (23.5°S–23.5°N) and southern extra-tropics (latitude > 23.5°S), the NDVI in 2015 is only moderately high (0–23.5°N) or around the multi-annual mean value (southern hemisphere).

Therefore, we conclude that, globally, 2015 is the greenest year of 2000–2015, in terms of both the mean annual NDVI value, and the number of grid cells where NDVI shows the highest rank in 2000–2015. This greenest signal is dominated by the extreme greenness in the growing season of the northern hemisphere, which has been examined in details in Bastos et al. (2017) and identified as a robust phenomenon independent of different satellite sensors used, or quality control procedures of the data. The Fig. S1 in Bastos et al. (2017) confirmed that both data from Terra and Aqua sensors show that 2015 has the highest growing season NDVI in 2000–2015. They also confirmed that such a conclusion is consistent among three quality control strategies of the Terra MODIS data used (Page 3, Bastos et al. 2017).

As the extreme greenness in 2015 is used as a starting point for our study and the main objective of our paper is to report the carbon dynamics and seasonal shifts in land carbon uptake associated with climate variations. We're fairly confident that sufficient evidences have been provided regarding the vegetation greenness for this specific year.

[Figure]

Fig. CS1 Seasonal distributions of land areas where 2015 shows the highest NDVI since 2000 as a function of latitude. Shaded areas represent different seasons stacked on top of each other.

[Figure]

Fig. CS2 (Top) Longitudinal distribution of the number of grid cells where 2015 NDVI ranks the highest in Q1 or Q4 of 2015 for the northern lands (latitude > 30°N). (Bottom) Spatial distribution of grid cells where NDVIs rank the highest for 2015 in Q1 or Q4 for the northern land (latitude > 30°N).

[Figure]

Fig. CS3 Annual NDVI anomalies for different latitudinal bands. The trend line is shown only for regions where significant simple linear regression over time is obtained.

**5. Are the results sufficient to support the interpretations and conclusions?**

Apart from the NDVI issues described above, the results are described in great detail and support the interpretation and conclusions.

[Response] Please refer to the responses to the comment above regarding the NDVI.

**6. Is the description of experiments and calculations sufficiently complete and precise to allow their reproduction by fellow scientists (traceability of results)?**

The calculations are mostly well described. The calculation of seasonal NDVI ranks seems to be a new approach to analyse NDVI time series (at least no reference is provided). Therefore I would recommend that the authors present some more details on this approach (at least in the Supplement) and ideally could provide also the code.

[Response] The NDVI ranking is mainly used to show the spatial distribution of abnormal greening in 2015, and that 2015 is in general the greenest year over 2000-2015 across the globe, which is dominated by extreme greening in northern land. Please also refer to our responses to the Comment 4 for more information. Fig. S3 in the revised Supplement shows the NDVI anomalies for different latitude bands, which clearly indicate that the highest annual NDVI over the globe is driven by the extreme green anomaly in the northern land (>23.5°N). The code used to generate Figure 1 in the texts is made available through a public repository (https://github.com/ChaoYue/ACPD-2016-1167).

**7. Do the authors give proper credit to related work and clearly indicate their own new/original contribution?**

Yes. The cited literature is relevant for this study. The own contributions of the authors are clear. However, I would recommend to provide a more detailed discussion on the link between vegetation greenness from satellites and carbon cycle or atmospheric $CO_2$ variability in order to improve the discussion section that is currently strongly focussed on the specificities of the year 2015. The results of this paper could be for example discussed with respect to the following relevant papers (Angert et al., 2005; Forkel et al., 2016; Gonsamo et al., 2017; Keenan et al., 2016; Myneni et al., 1997; Thomas et al., 2016).

[Response] We have substantially strengthened the discussion by making new analysis regarding the links among vegetation greenness, land carbon uptake anomalies and climate variations (Fig. 3, Fig. 4 in revised manuscript, Fig. S4, S5, S7 in the revised Supplement). Please also refer to our responses to the #2 Response to the #1 reviewer.

**8. Does the title clearly reflect the contents of the paper?**

Yes. However, I recommend to extent the analysis to more years to draw less specific concluisons for a single years. This might imply to change the title accordingly.

[Response] We extended the analysis by including more years and provided new figures in both main texts and the Supplement. Please also refer to our responses to the first and second comment by #1 reviewer. The manuscript title is also changed to: Vegetation greenness and land carbon flux anomaly associated with climate variations with a special focus on 2015.

**9. Does the abstract provide a concise and complete summary?**

Yes. The abstract is well written.

[Response] The abstract is updated to reflect the extra analysis conducted.

**10. Is the overall presentation well structured and clear?**

Yes.

**11. Is the language fluent and precise?**

Yes (as far as I can judge this). Some sentences are however too long and thus difficult to read, for example: lines 74-78, 90-93,

[Response] These sentences are re-phrased to enhance their readability.

**12. Are mathematical formulae, symbols, abbreviations, and units correctly defined and used?**

Yes. Units and proper axis descriptions are missing in Fig. 4.

[Response] This is fixed.

**13. Should any parts of the paper (text, formulae, figures, tables) be clarified, reduced, combined, or eliminated?**

Lines 71-74 are repeating lines 58-61 and can be merged.

[Response] Lines 58–61 are removed as they're repeated in section 2.1.1.

Lines 81-93: The affect of station network density on the inversion is well described for the CarboScope product. According to my understanding, the CAMS inversion should have the same problems. Please clarify how these issued are handled in the CAMS inversion.

[Response] The CAMS inversion uses sites with at least 5-year worth of data. It therefore has a denser (during the recent decade) but temporally evolving data coverage than Carboscope. The evolving network in CAMS causes changes in inverted CO2 fluxes that are superimposed on changes from biogeochemical drivers during the whole period.

Lines 131-139: Please make clear why the conversion from ppm to PgC was done and if there is any relevant uncertainty in this conversion factor.

[Response] The conversion of ppm to PgC is to express the atmospheric 'sink' of $CO_2$ in the same unit as carbon fluxes diagnosed from inversions, in order to coherently assess contributions from different fluxes to the AGR in 2015. This is explained in the revised texts. We used a conversion factor of 1ppm $CO_2$ = 2.12 Pg C (Ciais et al., 2014; Prather et al., 2012). For multi-decadal analysis, this ratio is correct given the sufficient mixing of $CO_2$ in the atmosphere because the value of 2.12 Pg C ppm$^{-1}$ considers the effect of a flux equilibrated with the troposphere (mixed in ≈ 1.2 years) and the stratosphere (mixed in ≈ 5 years). However we used this ratio on an annual basis, with the assumption that the entire atmosphere is well mixed within one year. This approximation is explicitly stated in the main text.

Such a ratio is mainly based on Ballantyne et al. (2012) & Le Quéré et al. (2016). We admit there are uncertainties in this ratio. Ballantyne et al. (2012) gave a relatively detailed discussion in their methods. On the one hand, the stratosphere is less well mixed with $CO_2$ than the troposphere, using $CO_2$ measurements at marine boundary layer (MBL) might overestimated the atmospheric $CO_2$ sink (thus implying a ratio that should be smaller than 2.12). But on the other hand, there is also $CO_2$ gradient from the continental boundary layer to marine boundary layer, which could compensate for the insufficient mixing in the stratosphere. Ballantyne et al. (2012) finally reached the conclusion that these two factors roughly cancel out each other by citing the close estimated MBL and whole atmosphere $CO_2$ concentrations. In our case, the partitioning of the AGR anomaly in 2015 is not our central purpose. The majority of conclusions reached by our analysis in the paper are based on the inversion-based land carbon uptake anomalies by the two inversion data sets used. Thus we argue the uncertainty of this conversion factor does not significantly impact our results.

Lines 163-164: What do you mean with "numerical instability"? Why could such an instability happen and why in 1993?

[Response] We mean rounding errors that accumulate rather than cancel. We have been experiencing these artifacts more often with increasing assimilation periods over the years, because the grid-point scale inversion problem becomes larger. To our knowledge, there is no particular reason why it happens in 1993

rather than in another year. We usually manage to remove these instabilities by re-running the inversion under slightly changed inversion configuration parameters, but this has not been done for this version.

Line 296: I thought that the Jena inversion system uses flat land prior fluxes. Are results from the LPJ model really used?

[Response] LPJ is used as a time-average spatial pattern, but concerning time variability as relevant here, the CarboScope prior is indeed flat (no prior interannual variations by periodical seasonal variations).

Figure 4: The figure could be much easier to read if you do some changes: #1 The red-green colour scale is not needed because the same information is already provided by the x-axis. Additionally, this colour scale might be not visible for colour-blind people. #2 The main purpose of this figure is to compare distributions of seasonal transitions from CAMS and Jena04. Overlaid histograms are not a good graphical choice. I would recommend to rather show distributions in terms of density lines, boxplots or violins which would make it easier to compare the distribution of CAMS and Jena04. The vertical lines for the year 2015 can be still added if you want to keep the focus on this year. #3 Please provide labels and units for the x-axis.

[Response] We removed the color in the vertical bars, and changed this plot into line plot of histograms for clarity. Labels and units are provided for x-axis.

**14. Are the number and quality of references appropriate?**

Yes, but also refer to my answer to the question #7.

[Response] We expanded substantially the discussion by citing relevant previous studies. The reference list is updated accordingly.

**15. Is the amount and quality of supplementary material appropriate?**

Yes, but an improved processing and uncertainty assessment of NDVI data might require more details in the supplementary material.

[Response] We provide further figures in the Supplement regarding the 2015 extreme greening.

**References**

Angert, A., Biraud, S., Bonfils, C., Henning, C. C., Buermann, W., Pinzon, J., Tucker, C. J. and Fung, I.: Drier summers cancel out the CO2 uptake enhancement induced by warmer springs, Proc. Natl. Acad. Sci. U. S. A., 102(31), 10823–7, doi:10.1073/pnas.0501647102, 2005.

D'Odorico, P., Gonsamo, A., Pinty, B., Gobron, N., Coops, N., Mendez, E. and Schaepman, M. E.: Intercomparison of fraction of absorbed photosynthetically active radiation products derived from satellite data over Europe, Remote Sens. Environ., 142, 141–154, doi:10.1016/j.rse.2013.12.005, 2014.

Fensholt, R. and Proud, S. R.: Evaluation of Earth Observation based global long term vegetation trends — Comparing GIMMS and MODIS global NDVI time series, Remote Sens. Environ., 119, 131–147, doi:10.1016/j.rse.2011.12.015, 2012.

Forkel, M., Carvalhais, N., Verbesselt, J., Mahecha, M., Neigh, C. and Reichstein, M.: Trend Change Detection in NDVI Time Series: Effects of Inter-Annual Variability and Methodology, Remote Sens., 5(5), 2113–2144, doi:10.3390/rs5052113, 2013.

Forkel, M., Carvalhais, N., Rödenbeck, C., Keeling, R., Heimann, M., Thonicke, K., Zaehle, S. and Reichstein, M.: Enhanced seasonal CO2 exchange caused by amplified plant productivity in northern ecosystems, Science, aac4971, doi:10.1126/science.aac4971, 2016.

Gonsamo, A., D'Odorico, P., Chen, J. M., Wu, C. and Buchmann, N.: Changes in vegetation phenology are not reflected in atmospheric CO2 and 13C/12C seasonality, Glob. Change Biol., n/a- n/a, doi:10.1111/gcb.13646, 2017.

Hird, J. N. and McDermid, G. J.: Noise reduction of NDVI time series: An empirical comparison of selected techniques, Remote Sens. Environ., 113(1), 248–258, doi:10.1016/j.rse.2008.09.003, 2009.

Holben, B. N.: Characteristics of maximum-value composite images from temporal AVHRR data, Int. J. Remote Sens., 7(11), 1417–1434, 1986.

Kandasamy, S., Baret, F., Verger, A., Neveux, P. and Weiss, M.: A comparison of methods for smoothing and gap filling time series of remote sensing observations – application to MODIS LAI products, Biogeosciences, 10(6), 4055–4071, doi:10.5194/bg-10-4055-2013, 2013.

Keenan, T. F., Prentice, I. C., Canadell, J. G., Williams, C. A., Wang, H., Raupach, M. and Collatz, G. J.: Recent pause in the growth rate of atmospheric CO2 due to enhanced terrestrial carbon uptake, Nat. Commun., 7, 13428, doi:10.1038/ncomms13428, 2016.

Kern, A., Marjanović, H. and Barcza, Z.: Evaluation of the Quality of NDVI3g Dataset against Collection 6 MODIS NDVI in Central Europe between 2000 and 2013, Remote Sens., 8(11), 955, doi:10.3390/rs8110955, 2016.

Myneni, R. B., Keeling, C. D., Tucker, C. J., Asrar, G. and Nemani, R. R.: Increased plant growth in the northern high latitudes from 1981 to 1991, Nature, 386(6626), 698–702, doi:10.1038/386698a0, 1997.

Scheftic, W., Zeng, X., Broxton, P. and Brunke, M.: Intercomparison of Seven NDVI Products over the United States and Mexico, Remote Sens., 6(2), 1057–1084, doi:10.3390/rs6021057, 2014.

Thomas, R. T., Prentice, I. C., Graven, H., Ciais, P., Fisher, J. B., Hayes, D. J., Huang, M., Huntzinger, D. N., Ito, A., Jain, A., Mao, J., Michalak, A. M., Peng, S., Poulter, B., Ricciuto, D. M., Shi, X., Schwalm, C., Tian, H. and Zeng, N.: CO2 and greening observations indicate increasing light-use efficiency in northern terrestrial ecosystems, Geophys. Res. Lett., doi:10.1002/2016GL070710, 2016.

References in the response:

Ballantyne, A. P., Alden, C. B., Miller, J. B., Tans, P. P. and White, J. W. C.: Increase in observed net carbon dioxide uptake by land and oceans during the past 50 years, Nature, 488(7409), 70–72, doi:10.1038/nature11299, 2012.

Bastos, A., Ciais, P., Park, T., Zscheischler, J., Yue, C., Barichivich, J., Myneni, R. B., Peng, S., Piao, S. and Zhu, Z.: Was the extreme Northern Hemisphere greening in 2015 predictable?, Environ. Res. Lett., 12(4), 044016, doi:10.1088/1748-9326/aa67b5, 2017.

Ciais, P., Sabine, C., Bala, G., Bopp, L., Brovkin, V., Canadell, J., Chhabra, A., DeFries, R., Galloway, J., Heimann, M. and others: Carbon and Other Biogeochemical Cycles, Clim. Change 2013 Phys. Sci. Basis Contrib. Work. Group Fifth Assess. Rep. Intergov. Panel Clim. Change, 465–570, 2014.

Gamon, J. A., Field, C. B., Goulden, M. L., Griffin, K. L., Hartley, A. E., Joel, G., Peñuelas, J. and Valentini, R.: Relationships Between NDVI, Canopy Structure, and Photosynthesis in Three Californian Vegetation Types, Ecol. Appl., 5(1), 28–41, doi:10.2307/1942049, 1995.

Ide, R., Nakaji, T. and Oguma, H.: Assessment of canopy photosynthetic capacity and estimation of GPP by using spectral vegetation indices and the light–response function in a larch forest, Agric. For. Meteorol., 150(3), 389–398, doi:10.1016/j.agrformet.2009.12.009, 2010.

Myneni, R. B., Keeling, C. D., Tucker, C. J., Asrar, G. and Nemani, R. R.: Increased plant growth in the northern high latitudes from 1981 to 1991, Nature, 386(6626), 698–702, doi:10.1038/386698a0, 1997.

Prather, M. J., Holmes, C. D. and Hsu, J.: Reactive greenhouse gas scenarios: Systematic exploration of uncertainties and the role of atmospheric chemistry, Geophys. Res. Lett., 39(9), L09803, doi:10.1029/2012GL051440, 2012.

Le Quéré, C., Andrew, R. M., Canadell, J. G., Sitch, S., Korsbakken, J. I., Peters, G. P., Manning, A. C., Boden, T. A., Tans, P. P., Houghton, R. A., Keeling, R. F., Alin, S., Andrews, O. D., Anthoni, P., Barbero, L., Bopp, L., Chevallier, F., Chini, L. P., Ciais, P., Currie, K., Delire, C., Doney, S. C., Friedlingstein, P., Gkritzalis, T., Harris, I., Hauck, J., Haverd, V., Hoppema, M., Klein Goldewijk, K., Jain, A. K., Kato, E., Körtzinger, A., Landschützer, P., Lefèvre, N., Lenton, A., Lienert, S., Lombardozzi, D., Melton, J. R., Metzl, N., Millero, F., Monteiro, P. M. S., Munro, D. R., Nabel, J. E. M. S., Nakaoka, S., O'Brien, K., Olsen, A., Omar, A. M., Ono, T., Pierrot, D., Poulter, B., Rödenbeck, C., Salisbury, J., Schuster, U., Schwinger, J., Séférian, R., Skjelvan, I., Stocker, B. D., Sutton, A. J., Takahashi, T., Tian, H., Tilbrook, B., Laan-Luijkx, I. T. van der, Werf, G. R. van der, Viovy, N., Walker, A. P., Wiltshire, A. J. and Zaehle, S.: Global Carbon Budget 2016, Earth Syst. Sci. Data, 8(2), 605–649, doi:10.5194/essd-8-605-2016, 2016.

Tanja, S., Berninger, F., Vesala, T., Markkanen, T., Hari, P., Mäkelä, A., Ilvesniemi, H., Hänninen, H., Nikinmaa, E., Huttula, T., Laurila, T., Aurela, M., Grelle, A., Lindroth, A., Arneth, A., Shibistova, O. and Lloyd, J.: Air temperature triggers the recovery of evergreen boreal forest photosynthesis in spring, Glob. Change Biol., 9(10), 1410–1426, doi:10.1046/j.1365-2486.2003.00597.x, 2003.

Zhao, M. and Running, S. W.: Drought-Induced Reduction in Global Terrestrial Net Primary Production from 2000 Through 2009, Science, 329(5994), 940–943, doi:10.1126/science.1192666, 2010.

---

## Author Response (AR2)

The authors carefully revised the manuscript in response to the previous reviewer comments. Data issues and analysis approaches were well examined. Thus the results seem to be robust. However, I believe that this study is still not a substantial contribution to our understanding of global vegetation/Earth system dynamics. As the manuscript has a strong focus on the state of NDVI and land carbon exchange in the year 2015, it is a very particular description of the 2015 situation. This particular focus was already criticised during the previous stage of the review. The authors slightly expanded the focus in the revised version but decided to still keep the strong focus on the year 2015. Although I think that this focus does not make the manuscript very interesting for the scientific community, I can okay it. As the results are robust, I suggest to accept the manuscript as it is and to let the scientific community decide about the scientific significance of these results.

[Response] We appreciate the reviewer's efforts to review our manuscript and the general positive view on our revised manuscript. We have in fact used the year 2015 as an illustrative natural experiment to examine the responses of land ecosystems to a combination of extreme greening and the occurrence of a very strong El Niño event. We strived to make a balance between general analysis of land carbon dynamics and vegetation greenness and climate variations, by using all years when data are available, and a focus on the special case of 2015. Throughout the result and discussion sections, the interpretation of carbon dynamics of 2015 is the assumed focus but it is always put into a context of all historical years, since the "abnormal" character of this year can only be understood from comparison with other years. The focus on the year of 2015 as a special case to study carbon cycle responses to ENSO events does not make the current work less interesting to the scientific community. In fact, on the most recent 10th International Carbon Dioxide Conference in Switzerland, there was a full session dedicated to the 2015 ENSO event

(https://www.conftool.com/icdc10/index.php?page=browseSessions&presentations=hi de), demonstrating the scientific community's interest on its influences on the earth system. Therefore, we contend that our study can come as a timely one and could be of interest to the community.

There are two key points in this manuscript that we hope can bring valuable insights to the scientific community. (1) It is almost taken granted that growing-season vegetation greenness in the northern hemisphere leads to land carbon sink, while our study highlights the importance of carbon dynamics at seasonal time scale. The decoupling of greening and land carbon uptake outside the growing season, and the enhanced autumn release due to warming or an intrinsic coupling of enhanced spring-summer uptake and autumn release, can actually offset the greening-induced carbon sink during the growing season. (2) We identified an extremely large transition to a carbon source (the largest over the data record of 1981–2015) in the tropics and southern hemisphere from the third to the fourth season, concomitant with the development of the strong El Niño in 2015. This finding highlights the potential future impact on land sink by the projected increase in the frequency of extreme El Niño events caused by the anthropogenic climate change.

In line with the suggestions made by the 2nd reviewer, we have revised the sections of abstract, introduction, discussion and conclusion in the updated version of the manuscript, to make clearer the scientific implications of our findings and to guide the

readers better interpret the results in the manuscript. The two points aforementioned are also described more clearly in the revised manuscript. All revised texts in the manuscript are tracked in blue.

**Anonymous Referee #2**

The authors have made very significant changes to the paper, and have addressed the reviewer comments in a relatively profound way. The article contains a very detailed and rich analysis, and I think that the balance between the year 2015 and other years is now adequate. The authors also improved the discussions of links to previous studies on the relations between the carbon cycle, vegetation greenness and climate anomalies. The scientific methods are very well documented and appear reasonable to me, and the results support the author's conclusions.

There is one aspect that I think the authors did not address so well, which is to work out a clear motivation for their study and a message that emerges from their results. As a meticulous case study for the year 2015, I think that the paper can be useful for other scientists in the same field. Having said that, the authors may reconsider if the visibility of their paper might be improved by some additional changes, especially in the abstract, introduction, and conclusions section.

To me, the fact that greening and carbon loss can occur together due to different seasons, locations and processes, invokes the question what the future trends of these changes are going to be. For example, would an increased ENSO frequency or amplitude have the potential to overcompensate the carbon uptake due to warming in the extratropics? Can the author's results be used for a simple back-of-the envelope calculation (for example, an extrapolation of their results into the future) to estimate a range of possibilities? Which sources of uncertainty in particular should be reduced in order to get a better estimate? Can the increased respiration in autumn in mid-latitudes overcompensate the increased uptake in spring, or does this only reflect the larger annual cycle with increasing overall uptake?

I understand that these may be questions too general and complex to answer. But my point is that using such kinds of questions at least as an orientation to guide the reader through the manuscript can improve its readability by pointing out the benefits that readers would have from spending time on the article. The aspects above are implicitly (and sometimes explicitly) already mentioned in the manuscript, but they are hidden in a large haystack of details. The very long and detailed documentation of results still leaves me a bit confused about how to link these results to the big questions about the carbon cycle. I believe that the authors have the expertise to work out these links in a more explicit way, without the need for much extra analysis.

[Response] We thank the reviewer's effort to review our manuscript and for the general appreciation of our efforts to revise the manuscript. In this updated version of the manuscript, we focused on revising the sections of abstract, introduction, discussion and conclusion, according to the reviewer's suggestions to make clearer the motivations of our study, and to guide the readers in interpreting the findings. The reviewers have given a few questions as examples that are relevant with our research objectives. Our revisions of the introduction and abstract are centered on these and other similar

questions. Please refer to the updated manuscript. All revised texts in the manuscript are tracked in blue.

**Minor comments:**

- The overall language of the article is good, but I would recommend that a native speaker takes a final look. There are some issues with missing articles especially. examples:

[Response] All following suggested language corrections below are accepted. The language of the manuscript has been improved by professional copy-editing service.

line 482: outside the growing season, line 486: a strong transition to a carbon source, line 487: observations showing a strong dependence of... line 558: The validity period of ... is 2004-2015, although data for the whole time span is available. line 785: still not good English – should read "apparent paradox" line 859: more common would be "linked to", or "associated with" line 969: tropical vegetation shows line 1034: decoupled outside the growing season ... The transition into a carbon source

- lines 559-562: I actually like the explanation in the response to my previous review more than the version in the article because it really eplains what the validity period is and why the estimates deteriorate outside. Readers who are less familiar with these technical details might not understand this paragraph as it is.

[Response] We replaced the relevant text with the explanations provided as the response to previous review comments and hope it is now more clear.

- line 683: The reason to remove year 1993 seems to be that it is an outlier and the authors conclude that the method fails in this case. But what assures the authors that a similar problem does not occur in other years as well, with a smaller effect on the results?

[Response] To some extent yes. The Jena inversion shows not such extreme transition as is seen in the CAMS inversion in 1993. In the case of strong negative transitions in the tropics between Q3 and Q4 in 2015, both CAMS and Jena show consistent transitions and the strong negative transitions are associated with a lower-than-normal annual land carbon sink in 2015. The year 1993, however, shows a reasonably strong annual carbon sink by the CAMS inversion and thus a negative carbon transition of -2.85 Pg C lower than  $-4\sigma$  is considered as an outlier.

- line 822: What is meant with a "first-order difference"? Couldn't one just remove "first-order"?

[Response] We removed "first-order" here to remove the unnecessary complexity.

**- line 1041: What is meant with opposing ENSO events?**

[Response] We mean opposing ENSO phases (i.e., the cold phase of La Nina versus the warm phase of El Niño). This is now specified in the text.

- line 1044: I am still quite sceptical about the use of the term "abrupt" in this context. I

would prefer the term "large climate anomalies". An abrupt shift usually implies consequences that are permanent.

[Response] We use "abrupt" here to mean more of "extreme" shift, such as the shift from an extremely low precipitation anomaly in Q3 to an extremely high one in Q4 for the region of temperate Northern Hemisphere (TeNH) as shown in Fig. S5g–h. We replaced "abrupt" by "extreme".

- line 1073-1075: This statement goes into the direction that I am driving at above and should be explained a bit more. It almost reads like an afterthought, but could be a major argument for the purpose of this study in the introduction. Why is this study an interesting test bed, and how? And after having done the analysis, how should one proceed to evaluate the models and gain understanding? Hence, the authors may want to fill this interesting statement with some more flavour and serve it as an appetizer instead of a dessert.

[Response] We have revised the sections of abstract, introduction, discussion and conclusion and have use questions as such proposed by the reviewer to lay down a concrete background for our study and to guide the readers to interpret our findings. Please refer to the revisions made in these sections.

[revised manuscript text omitted]